# Stable Neural Stochastic Differential Equations in Analyzing Irregular Time Series Data

**YongKyung Oh**[*], **Dong-Young Lim**[*], & **Sungil Kim**[†]
Ulsan National Institute of Science and Technology, Republic of Korea
`{yongkyungoh, dlim, sungil.kim}@unist.ac.kr`

## Abstract

Irregular sampling intervals and missing values in real-world time series data present challenges for conventional methods that assume consistent intervals and complete data. Neural Ordinary Differential Equations (Neural ODEs) offer an alternative approach, utilizing neural networks combined with ODE solvers to learn continuous latent representations through parameterized vector fields. Neural Stochastic Differential Equations (Neural SDEs) extend Neural ODEs by incorporating a diffusion term, although this addition is not trivial, particularly when addressing irregular intervals and missing values. Consequently, careful design of drift and diffusion functions is crucial for maintaining stability and enhancing performance, while incautious choices can result in adverse properties such as the absence of strong solutions, stochastic destabilization, or unstable Euler discretizations, significantly affecting Neural SDEs' performance. In this study, we propose three stable classes of Neural SDEs: Langevin-type SDE, Linear Noise SDE, and Geometric SDE. Then, we rigorously demonstrate their robustness in maintaining excellent performance under distribution shift, while effectively preventing overfitting. To assess the effectiveness of our approach, we conduct extensive experiments on four benchmark datasets for interpolation, forecasting, and classification tasks, and analyze the robustness of our methods with 30 public datasets under different missing rates. Our results demonstrate the efficacy of the proposed method in handling real-world irregular time series data.

## 1 Introduction

Conventional deep learning models, such as Recurrent Neural Network (RNN) (Rumelhart et al., 1986; Medsker & Jain, 1999), Long Short-term Memory (LSTM) (Hochreiter & Schmidhuber, 1997) and Gated Recurrent Unit (GRU) (Chung et al., 2014), consider time series data as consecutive discrete subsets, struggling with irregularly-sampled or partially-observed data (Mozer et al., 2017). To better reflect the underlying continuous process of the time series, researchers have proposed Neural Differential Equation (NDE)-based methods that enable deep learning models to learn continuous-time dynamics and the underlying temporal structure (Chen et al., 2018; Tzen & Raginsky, 2019; Jia & Benson, 2019; Liu et al., 2019; Kidger et al., 2020; 2021a;b; Ansari et al., 2023).

Unlike discrete representations from the conventional methods, *neural ordinary differential equations* (Neural ODEs) (Chen et al., 2018) directly learn continuous latent representation (or latent state) based on a vector field parameterized by a neural network. Kidger et al. (2020) introduced *neural controlled differential equations* (Neural CDEs), which are continuous-time analogs of RNNs that employ controlled paths to represent irregular time series.

As an extension of Neural ODEs, *neural stochastic differential equations* (Neural SDEs) have been introduced with a focus on aspects such as gradient computation, variational inference for latent spaces, and uncertainty quantification. For example, Tzen & Raginsky (2019) considers a variation inference for the diffusion limit in deep latent Gaussian models. Li et al. (2020) develops reverse-mode automatic differentiation for solutions of SDEs. Neural SDEs that incorporate various regularization

---

[*]These two authors are equal contributors to this work and designated as co-first authors.
[†]Corresponding Author

mechanisms such as dropout and noise injection were introduced in Liu et al. (2019). Kong et al. (2020) utilized a framework of Neural SDEs to quantify uncertainties in deep neural networks.

Despite advancements in the literature, there still exist unresolved issues and a lack of understanding in the modeling and robustness of Neural SDEs under *distribution shift*, motivating us to extend Neural SDE approaches into two directions. Firstly, as illustrated in our motivating example, the naïve implementation of Neural SDEs often fails to train. This is due to the fact that the behavior of the solution to SDEs can vary significantly depending on drift and diffusion functions, emphasizing the need for a study on an optimal selection of drift and diffusion functions. Secondly, irregular time series data, due to the nature of time variables, irregularity, and missingness, is prone to experiencing distribution shifts, which requires a substantial demand on model robustness (Li et al., 2021; Zhou et al., 2023).

The performance of deep learning models often deteriorates when applied to data distributions that are different from the original training distributions (Ovadia et al., 2019). This vulnerability under distribution shift[1] has been widely observed across various domains including healthcare, computer vision, and NLP (Leek et al., 2010; Bandi et al., 2018; Lazaridou et al., 2021; Miller et al., 2020; 2021). Therefore, theoretical guarantees to maintain robustness against distribution shift is a critical concern within modern deep learning communities. However, to the best of our knowledge, it has yet been explored to address this specific issue in the context of Neural SDEs.

To address these research gaps, this study introduces three classes of Neural SDEs to capture complex dynamics and improve robustness under distribution shift in time series data. While the drift and diffusion terms in the existing Neural SDEs are directly approximated by neural networks, the proposed Neural SDEs are trained based on theoretically well-defined SDEs. As a result, the new framework achieves state-of-the-art results in extensive experiments.

The main contributions of this study are summarized as follows. Firstly, we present three stable classes of Neural SDEs based on Langevin-type SDE, Linear Noise SDE, and Geometric SDE, and then combine the concept of controlled paths into the drift term of the proposed Neural SDEs to effectively capture sequential observations. Secondly, we show the existence and uniqueness of the solutions of these SDEs. In particular, we show that the Neural Geometric SDE shares properties with deep `ReLU` networks. Thirdly, we theoretically demonstrate that our proposed Neural SDEs maintain excellent performance under distribution shift due to missing data, while effectively preventing overfitting. Lastly, we conduct extensive experiments on four benchmark datasets for interpolation, forecasting, and classification tasks, and analyze the robustness of our methods with 30 public datasets under different missing rates. The proposed Neural SDEs show not only powerful empirical performance in terms of various measures but also remain robust to missing data.

## 2 RELATED WORK & PRELIMINARIES

**Notations.** Let $(\Omega, \mathcal{F}, P)$ be a probability space. Fix integer $d \geq 1$. For an $\mathbb{R}^d$-valued random variable $X$, its law is denoted by $\mathcal{L}(X)$. $|\cdot|$ is the Euclidean norm where the dimension of the space may vary depending on the context. We denote by $L^p(\Omega)$ the set of $X$ with $(\mathbb{E}[|X|^p])^{1/p} < \infty$. Let $\mathcal{P}(\mathbb{R}^d)$ denote the set of probability measures on $\mathcal{B}(\mathbb{R}^d)$. For two probability measures $\mu, \nu \in \mathcal{P}(\mathbb{R}^d)$, let $\mathcal{C}(\mu, \nu)$ denote the set of probability measures $\Pi$ on $\mathcal{B}(\mathbb{R}^{2d})$ such that its respective marginals are $\mu$ and $\nu$. Then, the Wasserstein distance of order $p \geq 1$ is defined as

$$\mathcal{W}_p(\mu, \nu) = \inf_{\Pi \in \mathcal{C}(\mu, \nu)} \left( \int_{\mathbb{R}^d} \int_{\mathbb{R}^d} |x - x'|^p \Pi(\mathrm{d}x, \mathrm{d}x') \right)^{1/p}.$$

### 2.1 NEURAL DIFFERENTIAL EQUATION METHODS

**Neural ODEs.** Let $\boldsymbol{x} \in \mathbb{R}^{d_x}$ denote the input data where $d_x$ is its dimension. Consider a latent representation $\boldsymbol{z}(t) \in \mathbb{R}^{d_z}$ at time $t$, which is given by

$$\boldsymbol{z}(t) = \boldsymbol{z}(0) + \int_0^t f(s, \boldsymbol{z}(s); \theta_f) \mathrm{d}s \quad \text{with } \boldsymbol{z}(0) = h(\boldsymbol{x}; \theta_h), \tag{1}$$

---

[1]This phenomenon occurs when the statistical properties of the target distribution that the model aims to predict shift over time, leading to discrepancies between the distributions of training data and test data.

where $h : \mathbb{R}^{d_x} \to \mathbb{R}^{d_z}$ is an affine function with parameter $\theta_h$. In Neural ODEs, $f(t, \boldsymbol{z}(t); \theta_f)$ is a neural network parameterized by $\theta_f$ to approximate $\frac{\mathrm{d}\boldsymbol{z}(t)}{\mathrm{d}t}$. To solve the integral problem in Equation (1), Neural ODEs rely on ODE solvers, such as the explicit Euler method (Chen et al., 2018). Since we can freely choose the upper limit $t$ of the integration, we can predict $\boldsymbol{z}$ at any time $t$. That is, once $h(\cdot; \theta_h)$ and $f(\cdot, \cdot; \theta_f)$ have been learned, then we are able to compute $\boldsymbol{z}(t)$ for any $t \geq 0$. Note that the extracted features $\boldsymbol{z}(t)$ are used for various tasks such as classification and regression. However, the solution to a Neural ODE is determined by its initial condition, making it inadequate for incorporating incoming information into a differential equation.

**Neural CDEs.** To address this issue, Kidger et al. (2020) proposed Neural CDEs as a continuous analogue of RNNs by combining a controlled path $X(t)$ of the underlying time-series data with Neural ODEs. Specifically, given the sequential data $\boldsymbol{x} = (x_0, x_1, \ldots, x_n)$, $\boldsymbol{z}(t)$ is determined by

$$\boldsymbol{z}(t) = \boldsymbol{z}(0) + \int_0^t f(s, \boldsymbol{z}(s); \theta_f) \mathrm{d}X(s) \quad \text{with } \boldsymbol{z}(0) = h(x_0; \theta_h), \tag{2}$$

where the integral is the Riemann–Stieltjes integral and $X(t)$ is chosen as a natural cubic spline path (Kidger et al., 2020) or hermite cubic splines with backward differences (Morrill et al., 2022) of the underlying time-series data. Differently from Neural ODEs, $f(t, \boldsymbol{z}(t); \theta_f)$, called CDE function, is a neural network parameterized by $\theta_f$ to approximate $\frac{\mathrm{d}\boldsymbol{z}(t)}{\mathrm{d}X(t)}$. The integral problem in Equation (2) can be solved by using existing ODE solvers since $\frac{\mathrm{d}\boldsymbol{z}(t)}{\mathrm{d}t} = f(t, \boldsymbol{z}(t); \theta_f) \frac{\mathrm{d}X(t)}{\mathrm{d}t}$.

**Neural SDEs.** Neural SDEs are an extension of Neural ODEs, which allows for describing the stochastic evolution of sample paths, rather than the deterministic evolution (Han et al., 2017; Tzen & Raginsky, 2019; Jia & Benson, 2019; Liu et al., 2019; Kidger et al., 2021a;b; Park et al., 2021). The latent representation $\boldsymbol{z}(t)$ of Neural SDEs is governed by the following SDE:

$$\boldsymbol{z}(t) = \boldsymbol{z}(0) + \int_0^t f(s, \boldsymbol{z}(s); \theta_f) \mathrm{d}s + \int_0^t g(s, \boldsymbol{z}(s); \theta_g) \mathrm{d}W(s) \quad \text{with } \boldsymbol{z}(0) = h(\boldsymbol{x}; \theta_h), \tag{3}$$

where $\{W_t\}_{t \geq 0}$ is a $d_z$-dimensional Brownian motion, $f(\cdot, \cdot; \theta_f)$ is the drift function, $g(\cdot, \cdot; \theta_g)$ is the diffusion function and the second integral on the right-hand side is the Itô integral. Neural SDEs assume that drift and diffusion functions are represented by neural networks.

## 2.2 LIMITATIONS OF NAÏVE NEURAL SDEs

The naïve implementation of Neural SDEs without sufficient consideration can lead to adverse results such as the absence of a unique strong solution (Mao, 2007; Øksendal & Øksendal, 2003), stochastic destabilization (Mao, 1994; Appleby et al., 2008), and/or unstable Euler discretization[2] (Hutzenthaler et al., 2011; Roberts & Tweedie, 1996). Therefore, careful design of the diffusion term is essential to improve the stability and efficacy of Neural SDEs.

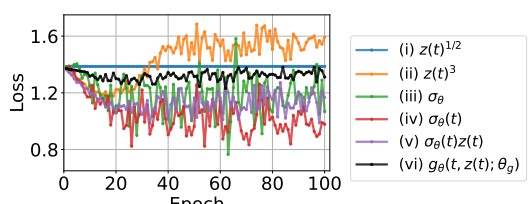

Figure 1: Comparison of test losses for Neural SDEs with six different diffusion functions on the 'BasicMotions' dataset at a 50% missing rate

The significance of a well-defined diffusion term becomes evident in the following illustrative example. Figure 1 displays a comparison of test losses for Neural SDEs employing six distinct diffusion functions $g(t, \boldsymbol{z}(t); \theta_g)$: (i) $\sqrt{\boldsymbol{z}(t)}$ (Hölder continuity with the exponent $1/2$), (ii) $|\boldsymbol{z}(t)|^3$ (non-Lipschitz continuity), (iii) $\sigma_\theta$ (constant), (iv) time-dependent function $\sigma(t; \theta_\sigma)$ (additive noise), (v) time-dependent function $\sigma(t; \theta_\sigma)\boldsymbol{z}(t)$ (multiplicative noise), and (vi) neural network $g_\theta(t, \boldsymbol{z}(t); \theta_g)$ (naïve Neural SDE). As depicted in the figure, the performance of Neural SDEs can vary significantly depending on the choice of diffusion functions. In particular, we observe that the non-Lipschitz continuous diffusion function $|\boldsymbol{z}(t)|^3$ and the naïve diffusion function are inadequate for stabilizing Neural SDEs. However, surprisingly, the influence of drift function and diffusion function design on the performance of Neural SDEs has been underexplored and remains poorly understood.

---

[2]The numerical approximation of Euler discretization for SDEs may explode in a finite time.

## 3 METHODOLOGY

In this section, our objective is to explore the characteristics of well-designed Neural SDEs. To achieve this, we propose three distinct classes of Neural SDEs, each with unique drift and diffusion functions, and conduct a comprehensive investigation of their theoretical properties. These three classes of Neural SDEs include Neural Langevin-type SDE (LSDE), Neural Linear Noise SDE (LNSDE), and Neural Geometric SDE (GSDE).

### 3.1 THE PROPOSED NEURAL SDES

**Langevin-type SDE.** Neural LSDE, motivated by a class of Langevin SDEs, is defined by

$$d\boldsymbol{z}(t) = \gamma(\boldsymbol{z}(t); \theta_\gamma)dt + \sigma(t; \theta_\sigma)dW(t) \quad \text{with } \boldsymbol{z}(0) = h(\boldsymbol{x}; \theta_h), \tag{4}$$

where the initial condition $h(\cdot; \theta_h)$ is an affine function with parameter $\theta_h$, the drift function $\gamma(\cdot; \theta_\gamma)$ is a neural network with parameter $\theta_\gamma$ and the diffusion function $\sigma(\cdot; \theta_\sigma)$ is a neural network with parameter $\theta_\sigma$. Note that the drift function of Neural LSDE is not explicitly dependent on time. While the Langevin SDE is a popular tool in recent years for Markov Chain Monte Carlo (MCMC) and stochastic optimization, it has been never explored in the context of Neural SDEs.

Assume that there exists $U$ such that $\nabla U(x) = \gamma(x; \theta_\gamma)$ for all $x \in \mathbb{R}^{d_z}$ and $\lim_{t\to\infty} \sigma(t; \theta_\sigma) = \sigma_0$ for some positive constant $\sigma_0$. Then, it is well known that, under mild conditions[3], the Langevin SDE in Equation (4) admits a unique invariant measure $\pi$, which is indeed the Gibbs measure $\pi(x) \propto \exp\left(\frac{2U(x)}{\sigma_0^2}\right)$ (T.-S. Chiang & Sheu, 1987; Raginsky et al., 2017). The existence of a unique invariant measure of the Langevin SDEs may be suitable for illustrating the phenomenon where the distributions of latent state values become invariant as the depth of the layers increases.

**Linear Noise SDE.** Neural LNSDE is governed by the following SDE with linear multiplicative noise:

$$d\boldsymbol{z}(t) = \gamma(t, \boldsymbol{z}(t); \theta_\gamma)dt + \sigma(t; \theta_\sigma)\boldsymbol{z}(t)dW(t) \quad \text{with } \boldsymbol{z}(0) = h(\boldsymbol{x}; \theta_h), \tag{5}$$

where the initial condition $h(\cdot; \theta_h)$ is an affine function with parameter $\theta_h$, the drift function $\gamma(\cdot, \cdot; \theta_\gamma)$ is a neural network with parameter $\theta_\gamma$ and the diffusion function $\sigma(\cdot; \theta_\sigma)$ is a neural network with parameter $\theta_\sigma$. The form of Neural LNSDE was also proposed in Liu et al. (2019), which merely reiterates the well-known results for stability such as exponential stability. In contrast, our work provides a novel theoretical analysis on the robustness of the trained model under drift shift, a crucial topic in recent deep learning research.

**Geometric SDE.** Neural GSDE considers the following SDE:

$$\frac{d\boldsymbol{z}(t)}{\boldsymbol{z}(t)} = \gamma(t, \boldsymbol{z}(t); \theta_\gamma)dt + \sigma(t; \theta_\sigma)dW(t) \quad \text{with } \boldsymbol{z}(0) = h(\boldsymbol{x}; \theta_h), \tag{6}$$

where the initial condition $h(\cdot; \theta_h)$ is an affine function with parameter $\theta_h$, the drift function $\gamma(\cdot, \cdot; \theta_\gamma)$ is a neural network with parameter $\theta_\gamma$ and the diffusion function $\sigma(\cdot; \theta_\sigma)$ is a neural network with parameter $\theta_\sigma$. Due to its exponential form, Neural GSDE has distinct characteristics associated with deep `ReLU` networks such as a unique positive solution and an absorbing state of $0$, which are not observed in Neural LSDE and Neural LNSDE (see Proposition 3.4 (ii)).

### 3.2 PROPERTIES OF THE PROPOSED NEURAL SDES

We impose some assumptions on neural networks used in Neural SDEs as follows.

**Assumption 3.1.** For any neural network $s(t, x; \theta_s) : \mathbb{R}_+ \times \mathbb{R}^{d_1} \to \mathbb{R}^{d_2}$ with parameter $\theta_s$, there exists a positive constant $L_s > 0$ such that for all $t \geq 0$ and $x, x' \in \mathbb{R}^{d_1}$

$$|s(t, x; \theta_s) - s(t, x'; \theta_s)| \leq L_s|x - x'|.$$

---

[3]For example, $U$ can be nonconvex and have superlinear gradients (Raginsky et al., 2017; Lim et al., 2023a).

Neural networks with activation functions such as `tanh`, `ReLU`, and `sigmoid` functions generally satisfy the Lipschitz continuity condition in Assumption 3.1 (Chen et al., 2018; Liu et al., 2019; Virmaux & Scaman, 2018; Fazlyab et al., 2019; Latorre et al., 2020).

**Assumption 3.2.** For any neural network $s(t, x; \theta_s) : \mathbb{R}_+ \times \mathbb{R}^{d_1} \to \mathbb{R}^{d_2}$ with parameter $\theta_s$ where the last layer is the `ReLU` or linear function, there exists a positive constant $K_s > 0$ such that

$$|s(t, x; \theta_s)| \leq K_s(1 + |x|), \quad \text{for all } t \geq 0, x \in \mathbb{R}^{d_1}.$$

On the other hand, when `tanh` function or `sigmoid` function is applied at the last layer, we have

$$|s(t, x; \theta_s)| \leq K_s, \quad \text{for all } t \geq 0, x \in \mathbb{R}^{d_1}. \tag{7}$$

Assumption 3.2 implies that neural networks satisfy the linear growth condition. Assumptions 3.1 and 3.2 are commonly imposed in the literature on NDE-based methods to ensure the existence and uniqueness of solutions for SDEs (Chen et al., 2018; Liu et al., 2019; Kong et al., 2020).

**Assumption 3.3.** For any neural network $s(t, x; \theta_s) : \mathbb{R}_+ \times \mathbb{R}^{d_1} \to \mathbb{R}^{d_2}$ with parameter $\theta_s$, there exist positive constants $m, b > 0$ such that for all $t \geq 0, x \in \mathbb{R}^{d_1}$

$$\langle s(t, x; \theta_s),\, x \rangle \leq -m|x|^2 + b.$$

The condition in Assumption 3.3 is a typical assumption necessary for ensuring the uniform boundedness of the solutions to SDEs and diffusion approximation (Mattingly et al., 2002; Raginsky et al., 2017; Xu et al., 2018; Zhang et al., 2017). We emphasize that we do not impose convexity on the drift and diffusion functions.

Under Assumptions 3.1 and 3.2, we show that SDEs in Equations (4), (5), and (6) have their unique strong solutions. In particular, the geometric SDE has interesting properties that are suitable for modeling deep `ReLU` networks.

**Proposition 3.4.** *Let Assumptions 3.1 and 3.2 hold. Then, we have*

*(i) The SDEs in Equation (4) and Equation (5) have their unique strong solutions.*

*(ii) Assume that the activation function in $\gamma$ is either a `tanh` or `sigmoid` function. Then, the SDE in Equation (6) has a unique strong solution $\boldsymbol{z}(t)$, which is nonnegative almost surely, i.e., $P(\boldsymbol{z}(t) \geq 0 \text{ for all } t \geq 0) = 1$ a.s. Furthermore, state 0 is an absorbing state.*

The proof for Proposition 3.4 can be found in Appendix A. Intuitively, Proposition 3.4 (ii) implies that the latent representation $\boldsymbol{z}(t)$ of the Neural GSDE has always nonnegative values. Furthermore, once the one of values of the latent representation reaches 0, it remains there forever. This property is consistent with that of `ReLU` networks which have positive values for activated neurons and zeros for deactivated neurons. In particular, the fact that deactivated neurons never turn to activated neurons corresponds to the property of state 0 being an absorbing state.

### 3.3 ROBUSTNESS UNDER DISTRIBUTION SHIFT

This subsection provides insight into why the proposed Neural SDEs remain robust to input perturbations, such as missing data, and prevent overfitting caused by differences between training and test datasets. To obtain our main results, we heavily rely on the analysis of stochastic stability. An overview of stochastic stability and relevant results are summarized in Appendix B.1.

Let $\boldsymbol{x}$ denote the input data, and its law be represented by $\mathcal{L}(\boldsymbol{x})$, i.e., $\boldsymbol{x}$ is a $\mathbb{R}^{d_x}$-valued random variable. Consider $\widetilde{\boldsymbol{x}}$ to be a perturbed version of the input data, with the law $\mathcal{L}(\widetilde{\boldsymbol{x}})$, such that

$$\sqrt{\mathbb{E}[|\boldsymbol{x} - \widetilde{\boldsymbol{x}}|^2]} \leq \rho, \tag{8}$$

where $\rho > 0$ represents the degree of distribution shift. Suppose that we consider the task of classification or regression for time series data and use a feed-forward neural network $F$ composed of two fully connected layers as follows:

$$y = F(\boldsymbol{z}(T); \theta_F), \quad \widetilde{y} = F(\widetilde{\boldsymbol{z}}(T); \theta_F), \tag{9}$$

where $y, \widetilde{y}$ represent the predictions from the extracted feature $\boldsymbol{z}(T)$ and its perturbed version $\widetilde{\boldsymbol{z}}(T)$, respectively. Note that $\boldsymbol{z}(T)$ and $\widetilde{\boldsymbol{z}}(T)$ are outputs of Neural SDEs with input data $\boldsymbol{x}$ and $\widetilde{\boldsymbol{x}}$, respectively. For our stability analysis, we assume $\sigma(t; \theta_\sigma)$ to be either a constant $\sigma_\theta$ or to have a limit such that $\lim_{t \to \infty} \sigma(t; \theta_\sigma) =: \sigma_\theta$.

**Theorem 3.5.** *(Robustness of Neural LSDE) Let Assumptions 3.1, 3.2 and 3.3 hold. Let $\boldsymbol{x} \in L^4(\Omega)$ and $\widetilde{\boldsymbol{x}} \in L^4(\Omega)$ denote the input data and its perturbed version satisfying Equation (8). For the outputs $\boldsymbol{y}$, $\widetilde{\boldsymbol{y}}$ of Neural LSDEs in Equation (9), we have*

$$\mathcal{W}_1(\mathcal{L}(\boldsymbol{y}), \mathcal{L}(\widetilde{\boldsymbol{y}})) \leq \sqrt{3} L_F L_h c_1 e^{-c_2 T} \sqrt{(5 + 2\mathbb{E}[|\boldsymbol{x}|^4] + 2\mathbb{E}[|\widetilde{\boldsymbol{x}}|^4])} \rho,$$

$$\mathcal{W}_2(\mathcal{L}(\boldsymbol{y}), \mathcal{L}(\widetilde{\boldsymbol{y}})) \leq \sqrt{2\sqrt{3} L_h c_1} L_F e^{-c_2 T/2} \left(5 + 2\mathbb{E}[|\boldsymbol{x}|^4] + 2\mathbb{E}[|\widetilde{\boldsymbol{x}}|^4]\right)^{1/4} \sqrt{\rho},$$

*where $L_F$ is the Lipschitz constant of the neural network $F$ defined in Equation (9), $L_h$ is the Lipschitz constant of the initial condition $h$ in Equation (4) and positive constants $c_1$, $c_2$ are independent of $T$.*

**Theorem 3.6.** *(Robustness of Neural LNSDE and Neural GSDE) Let Assumptions 3.1, 3.2 and 3.3 hold. Let $\boldsymbol{x} \in L^2(\Omega)$ and $\widetilde{\boldsymbol{x}} \in L^2(\Omega)$ denote the input data and its perturbed version satisfying Equation (8), and $\boldsymbol{y}$ and $\widetilde{\boldsymbol{y}}$ denote their outputs defined in Equation (9).*

(i) *(Neural LNSDE) For $|\sigma_\theta|^2 > 2L_\gamma$ where $L_\gamma$ is the Lipschitz constant of the neural network $\gamma$ and for sufficiently large $T$, we have*

$$\mathcal{W}_1(\mathcal{L}(\boldsymbol{y}), \mathcal{L}(\widetilde{\boldsymbol{y}})) \leq L_F \exp\{-(|\sigma_\theta|^2 - 2L_\gamma)T/2\}(1 + \rho).$$

(ii) *(Neural GSDE) For $|\sigma_\theta|^2 > 2K_\gamma$ where $K_\gamma$ is the upper bound of the neural network $\gamma$ defined in Equation (7), and for sufficiently large $T$, we have*

$$\mathcal{W}_1(\mathcal{L}(\boldsymbol{y}), \mathcal{L}(\widetilde{\boldsymbol{y}})) \leq L_F \exp\{-(|\sigma_\theta|^2 - 2K_\gamma)T/2\}(1 + \rho).$$

The proofs for Theorem 3.5 and Theorem 3.6 can be found in Appendix B.2 and B.3, respectively. Theorem 3.5 and Theorem 3.6 provide non-asymptotic upper bounds for the differences between the output distributions of the original input data and its perturbed version, influenced by the degree of distribution shift $\rho$ and the depth of the Neural SDEs $T$. Smaller perturbations and larger depths yield smaller differences. Thus, the proposed Neural SDEs do not undergo abrupt changes with respect to $\rho$, exhibiting robust performance even if the input distribution shifts. In contrast, the lack of such stability in other Neural SDEs can yield dramatically different solutions from small changes in the input data, causing dramatic performance degradation or overfitting. The experimental findings presented in Section 4 and Appendix D.4 provide empirical support for our theoretical results.

### 3.4 INCORPORATING A CONTROLLED PATH TO NEURAL SDEs

To further improve empirical performance and effectively capture sequential observations like time-series data, we propose $\overline{\boldsymbol{z}}(t)$ that incorporates a controlled path in a nonlinear way as follows:

$$\overline{\boldsymbol{z}}(t) = \zeta(t, \boldsymbol{z}(t), X(t); \theta_\zeta), \tag{10}$$

where $X(t)$ is the controlled path and $\zeta$ is a neural network parameterized by $\theta_\zeta$. Then, we replace $\boldsymbol{z}(t)$ in the drift functions of the proposed Neural SDEs with $\overline{\boldsymbol{z}}(t)$ as presented in Equation (10). The effectiveness of utilizing $\overline{\boldsymbol{z}}(t)$ is confirmed through an ablation study in Section 4.3.

*Remark* 3.7. When combined with $\overline{\boldsymbol{z}}(t)$, the proposed Neural SDEs, including Neural LSDE, Neural LNSDE, and Neural GSDE, have their unique strong solutions. We refer to Appendix A.1 for a detailed discussion and verification.

## 4 EXPERIMENTS

### 4.1 SUPERIOR PERFORMANCE WITH REGULAR AND IRREGULAR TIME SERIES DATA

In this section, we conducted interpolation and classification experiments and evaluated the proposed Neural SDEs using three real-world datasets: PhysioNet Mortality (Silva et al., 2012), PhysioNet Sepsis (Reyna et al., 2019), and Speech Commands (Warden, 2018). For forecasting experiments, we utilized the MuJoCo (Tassa et al., 2018) dataset, and the results are detailed in Appendix E.

**Datasets.** The PhysioNet Mortality dataset contains multivariate time series data from 37 variables of Intensive Care Unit (ICU) records, with irregular measurements taken within the first 48 hours of admission. For our interpolation experiments, we use all 4,000 labeled and 4,000 unlabeled instances.

The PhysioNet Sepsis dataset includes 40,335 ICU patient cases and 34 time-dependent variables, aiming to classify each case according to the sepsis-3 criteria. This dataset represents irregular time series data, as only 10% of the values are sampled with their respective timestamps for each patient. We considered two scenarios: one with observational intensity (OI) and one without. In the scenario with OI, the observation index is attached to every value in the time series.

The Speech Commands dataset is a one-second-long audio data recorded with 35 distinct spoken words with the background noise. To create a balanced classification problem, ten labels[4] were selected from 34,975 time-series samples. Each preprocessed sample, utilizing Mel-frequency cepstral coefficients, has a time-series length of 161 and a 20-dimensional input size.

**Experimental Protocols.** We followed the recommended pipeline for interpolation experiments with the PhysioNet Mortality dataset, as outlined in Shukla & Marlin (2021) and its corresponding GitHub repository[5]. For classification experiments involving the PhysioNet Sepsis and Speech Commands datasets, we adhered to the experimental protocols described in Kidger et al. (2020) and its GitHub repository[6]. Our experiments were performed using a server on Ubuntu 22.04 LTS, equipped with an Intel(R) Xeon(R) Gold 6242 CPU and six NVIDIA A100 40GB GPUs. The source code can be accessed at `https://github.com/yongkyung-oh/Stable-Neural-SDEs`.

**Models.** We considered a range of models, including state-of-the-art models for both interpolation and classification tasks. These are **RNN-based models** ( RNN (Rumelhart et al., 1986; Medsker & Jain, 1999), LSTM (Hochreiter & Schmidhuber, 1997), GRU (Chung et al., 2014), GRU-$\Delta t$, and GRU-D (Che et al., 2018)), **Attention-based models** (MTAN Shukla & Marlin (2021), and MIAM Lee et al. (2022)), **Neural ODEs:** Neural ODEs (Chen et al., 2018), GRU-ODE (De Brouwer et al., 2019), ODE-RNN (Rubanova et al., 2019), ODE-LSTM (Lechner & Hasani, 2020), Latent-ODE (Rubanova et al., 2019), Augmented-ODE (Dupont et al., 2019), and ACE-NODE (Jhin et al., 2021))), **Neural CDEs** (Neural CDE Kidger et al. (2020), Neural RDE (Morrill et al., 2021), ANCDE (Jhin et al., 2023b), EXIT (Jhin et al., 2022), and LEAP (Jhin et al., 2023a)), and **Neural SDEs** (Neural SDEs (Tzen & Raginsky, 2019), and Latent SDE Li et al. (2020)).

**Results.** The results of the interpolation and classification experiments are summarized in the following tables. We have highlighted **the best** and the second best methods in the result tables.

Table 1 compares interpolation performance from 50% to 90% observed values in the test dataset. The proposed Neural SDEs consistently outperform all benchmark models, including the state-of-the-art method mTAND-Full (MTAN encoder–MTAN decoder model), at all observed time point settings.

Table 1: Interpolation performance versus percent observed time points on PhysioNet Mortality

| Methods | Mean Squared Error ($\times 10^{-3}$) | | | | |
|---|---|---|---|---|---|
| RNN-VAE | 13.418 ± 0.008 | 12.594 ± 0.004 | 11.887 ± 0.005 | 11.133 ± 0.007 | 11.470 ± 0.006 |
| L-ODE-RNN | 8.132 ± 0.020 | 8.140 ± 0.018 | 8.171 ± 0.030 | 8.143 ± 0.025 | 8.402 ± 0.022 |
| L-ODE-ODE | 6.721 ± 0.109 | 6.816 ± 0.045 | 6.798 ± 0.143 | 6.850 ± 0.066 | 7.142 ± 0.066 |
| mTAND-Full | 4.139 ± 0.029 | 4.018 ± 0.048 | 4.157 ± 0.053 | 4.410 ± 0.149 | 4.798 ± 0.036 |
| LatentSDE | 8.862 ± 0.036 | 8.864 ± 0.058 | 8.686 ± 0.122 | 8.716 ± 0.032 | 8.435 ± 0.077 |
| **Neural SDE** | 8.592 ± 0.055 | 8.591 ± 0.052 | 8.540 ± 0.051 | 8.318 ± 0.010 | 8.252 ± 0.023 |
| **Neural LSDE** | **3.799 ± 0.055** | **3.584 ± 0.055** | 3.457 ± 0.078 | **3.262 ± 0.032** | **3.111 ± 0.076** |
| **Neural LNSDE** | 3.808 ± 0.078 | 3.617 ± 0.129 | **3.405 ± 0.089** | 3.269 ± 0.057 | 3.154 ± 0.084 |
| **Neural GSDE** | 3.824 ± 0.088 | 3.667 ± 0.079 | 3.493 ± 0.024 | 3.287 ± 0.070 | 3.118 ± 0.065 |
| **Observed %** | **50%** | **60%** | **70%** | **80%** | **90%** |

Tables 2 and 3 demonstrate that the proposed Neural SDEs consistently outperform all other methods in classification tasks. For PhysioNet Sepsis, we report AUROC rather than accuracy due to dataset imbalance. While CDE-based methods typically benefit from OI (Kidger et al., 2020), our models perform well even without OI. This indicates that the proposed Neural SDEs leverage the advantages of incorporating a controlled path in the drift function and introducing stochasticity through the diffusion function. In Table 3, we do not report the performance of the naïve Nerual SDE due to

---

[4]They include 'yes', 'no', 'up', 'down', 'left', 'right', 'on', 'off', 'stop', and 'go'.

[5]`https://github.com/reml-lab/mTAN`

[6]`https://github.com/patrick-kidger/NeuralCDE`

training instability. Note that the proposed methods are less memory-efficient compared to CDE-based methods, and we leave memory usage improvement for future research.

Table 2: AUROC and memory usage (in MB) on PhysioNet Sepsis

| Methods | Test AUROC | | Memory | |
|---|---|---|---|---|
| | OI | No OI | OI | No OI |
| GRU-$\Delta t$ | 0.878 ± 0.006 | 0.840 ± 0.007 | 837 | 826 |
| GRU-D | 0.871 ± 0.022 | 0.850 ± 0.013 | 889 | 878 |
| GRU-ODE | 0.852 ± 0.010 | 0.771 ± 0.024 | 454 | 273 |
| ODE-RNN | 0.874 ± 0.016 | 0.833 ± 0.020 | 696 | 686 |
| Latent-ODE | 0.787 ± 0.011 | 0.495 ± 0.002 | 133 | 126 |
| ACE-NODE | 0.804 ± 0.010 | 0.514 ± 0.003 | 194 | 218 |
| Neural CDE | 0.880 ± 0.006 | 0.776 ± 0.009 | 244 | 122 |
| ANCDE | 0.900 ± 0.002 | 0.823 ± 0.003 | 285 | 129 |
| Neural SDE | 0.799 ± 0.007 | 0.796 ± 0.006 | 368 | 240 |
| Neural LSDE | 0.909 ± 0.004 | 0.879 ± 0.008 | 373 | 436 |
| Neural LNSDE | **0.911 ± 0.002** | 0.881 ± 0.002 | 341 | 445 |
| Neural GSDE | 0.909 ± 0.001 | **0.884 ± 0.002** | 588 | 280 |

Table 3: Accuracy and memory usage (in MB) on Speech Commands

| | Methods | Test Accuracy | Memory |
|---|---|---|---|
| RNN-based | RNN | 0.197 ± 0.006 | 1905 |
| | LSTM | 0.684 ± 0.034 | 4080 |
| | GRU | 0.747 ± 0.050 | 4609 |
| | GRU-$\Delta t$ | 0.453 ± 0.313 | 1612 |
| | GRU-D | 0.346 ± 0.286 | 1717 |
| NDE-based | GRU-ODE | 0.487 ± 0.018 | 171 |
| | ODE-RNN | 0.678 ± 0.276 | 1472 |
| | Latent-ODE | 0.912 ± 0.006 | 2668 |
| | Augmented-ODE | 0.911 ± 0.008 | 2626 |
| | ACE-NODE | 0.911 ± 0.003 | 3046 |
| | Neural CDE | 0.898 ± 0.025 | 175 |
| | ANCDE | 0.807 ± 0.075 | 180 |
| | LEAP | 0.922 ± 0.002 | 391 |
| | Neural LSDE | **0.927 ± 0.004** | 1187 |
| | Neural LNSDE | 0.923 ± 0.001 | 1164 |
| | Neural GSDE | 0.913 ± 0.001 | 1565 |

## 4.2 ROBUSTNESS TO MISSING DATA

We evaluated the performance of the proposed Neural SDEs on 30 datasets, including 15 univariate and 15 multivariate datasets, from the University of East Anglia (UEA) and the University of California Riverside (UCR) Time Series Classification Repository (Bagnall et al., 2018). We assess our models under regular (0% missing rate) and three missing rate conditions (30%, 50%, and 70%), adhering to the protocol by Kidger et al. (2020). Details on datasets and protocols along with the implementation of benchmark methods are summarized in Appendices C and D. We compare the average classification performance of each method using average accuracy and average rank.

Table 4: Classification performance on 30 datasets with regular and three missing rates (The values within the parentheses indicate the average of 30 individual standard deviations.)

| Methods | Regular datasets | | Missing datasets (30%) | | Missing datasets (50%) | | Missing datasets (70%) | |
|---|---|---|---|---|---|---|---|---|
| | Accuracy | Rank | Accuracy | Rank | Accuracy | Rank | Accuracy | Rank |
| RNN | 0.582 (0.064) | 13.9 | 0.513 (0.087) | 15.3 | 0.485 (0.088) | 16.6 | 0.472 (0.072) | 15.6 |
| LSTM | 0.633 (0.053) | 11.2 | 0.595 (0.060) | 12.1 | 0.567 (0.061) | 12.9 | 0.558 (0.058) | 12.5 |
| GRU | 0.672 (0.059) | 8.3 | 0.621 (0.063) | 10.2 | 0.610 (0.055) | 10.3 | 0.597 (0.062) | 10.3 |
| GRU-$\Delta t$ | 0.641 (0.070) | 10.3 | 0.636 (0.066) | 8.9 | 0.634 (0.056) | 8.7 | 0.618 (0.065) | 10.2 |
| GRU-D | 0.648 (0.071) | 10.3 | 0.624 (0.075) | 10.4 | 0.611 (0.073) | 11.1 | 0.604 (0.067) | 10.8 |
| MTAN | 0.648 (0.080) | 12.0 | 0.618 (0.099) | 10.7 | 0.618 (0.091) | 10.1 | 0.607 (0.078) | 9.8 |
| MIAM | 0.623 (0.048) | 11.0 | 0.603 (0.066) | 11.1 | 0.589 (0.063) | 12.2 | 0.569 (0.056) | 12.3 |
| GRU-ODE | 0.671 (0.067) | 9.8 | 0.663 (0.064) | 9.5 | 0.666 (0.059) | 8.3 | 0.655 (0.062) | 7.8 |
| ODE-RNN | 0.658 (0.063) | 9.1 | 0.635 (0.064) | 9.3 | 0.636 (0.067) | 8.2 | 0.630 (0.055) | 8.5 |
| ODE-LSTM | 0.619 (0.063) | 11.4 | 0.584 (0.064) | 12.1 | 0.561 (0.065) | 13.3 | 0.530 (0.085) | 12.9 |
| Neural CDE | 0.709 (0.061) | 8.2 | 0.706 (0.073) | 6.4 | 0.696 (0.064) | 6.5 | 0.665 (0.072) | 7.6 |
| Neural RDE | 0.607 (0.071) | 13.9 | 0.514 (0.064) | 14.9 | 0.468 (0.068) | 15.2 | 0.415 (0.077) | 16.3 |
| ANCDE | 0.693 (0.067) | 7.8 | 0.687 (0.068) | 7.2 | 0.683 (0.078) | 6.9 | 0.655 (0.067) | 7.1 |
| EXIT | 0.636 (0.073) | 11.1 | 0.633 (0.078) | 10.2 | 0.616 (0.075) | 10.6 | 0.599 (0.075) | 11.1 |
| LEAP | 0.444 (0.068) | 15.2 | 0.401 (0.078) | 16.3 | 0.425 (0.073) | 14.9 | 0.414 (0.070) | 14.7 |
| Latent SDE | 0.456 (0.073) | 16.6 | 0.455 (0.073) | 15.4 | 0.455 (0.069) | 15.0 | 0.446 (0.066) | 15.1 |
| Neural SDE | 0.526 (0.068) | 13.4 | 0.508 (0.066) | 13.1 | 0.517 (0.058) | 13.2 | 0.512 (0.066) | 12.9 |
| Neural LSDE | 0.717 (0.056) | 5.6 | 0.690 (0.050) | 6.4 | 0.686 (0.051) | 6.1 | 0.682 (0.067) | 5.2 |
| Neural LNSDE | **0.727 (0.047)** | **5.4** | **0.723 (0.050)** | **5.0** | **0.717 (0.054)** | **4.3** | **0.703 (0.054)** | **4.2** |
| Neural GSDE | 0.716 (0.065) | 5.7 | 0.707 (0.069) | 5.3 | 0.698 (0.063) | 6.1 | 0.689 (0.056) | 5.3 |

Table 4 shows that the proposed Neural SDEs consistently achieve top-tier performance in accuracy and rank across different missing rates, maintaining stable accuracy even with increased missing rates. Conversely, traditional RNN-based methods, such as RNN, LSTM, and GRU, experience noticeable performance drops as the missing rate increases. Furthermore, the proposed methods converge more rapidly and achieve lower loss values than the naïve Neural SDE, as depicted in Figure 2. Particularly, when comparing Figure 2(a) with Figure 1, it becomes evident that the proposed methods are overcoming the stability limitations of the naïve Neural SDE.

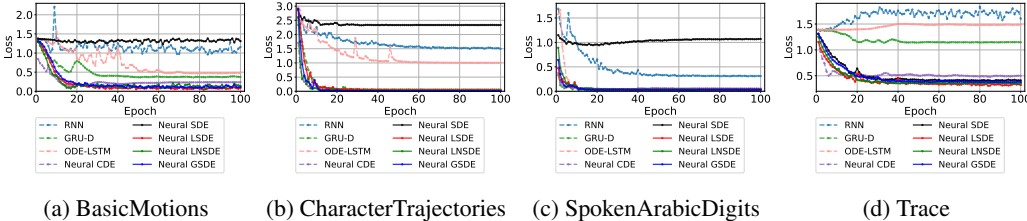

| (a) BasicMotions | (b) CharacterTrajectories | (c) SpokenArabicDigits | (d) Trace |

Figure 2: Comparing stability of test loss during model training with the four datasets at a 50% missing rate with the selected methods (Training 100 epochs without early-stopping.)

## 4.3 ABLATION STUDY

We conduct ablation experiments to show the performance impact of incorporating a controlled path into the drift function and using a nonlinear neural network for the diffusion function. Table 5 illustrates the effectiveness of a nonlinear neural network compared to an affine function in the diffusion function. The results reveal a fundamental performance difference between the three proposed Neural SDEs and the naïve Neural SDE, with the latter unable to bridge the gap by adopting a controlled path and a nonlinear diffusion function. The superior and consistent performance of our models across 15 univariate and 15 multivariate datasets underscores their robustness and generality. Ablation studies for the controlled path design and solver for Neural SDEs are described in Appendix D. Additionally, detailed results for each individual dataset are provided in Appendix F.

Table 5: Comparison of average accuracy and average cross-entropy loss from the ablation study ($\zeta$ indicates whether the drift function incorporates the controlled path or not. 'N' or 'L' denote the architecture of diffusion function $\sigma$ with an affine function or a nonlinear neural network.)

| Methods | $\zeta$ | $\sigma$ | Univariate datasets (15) | | Multivariate datasets (15) | | All datasets (30) | |
|---|---|---|---|---|---|---|---|---|
| | | | Accuracy | Loss | Accuracy | Loss | Accuracy | Loss |
| **Neural SDE** | O | N | 0.615 (0.090) | 0.736 (0.089) | 0.760 (0.036) | 0.618 (0.092) | 0.688 (0.063) | 0.677 (0.091) |
| | | L | 0.535 (0.095) | 0.838 (0.094) | 0.752 (0.040) | 0.633 (0.087) | 0.643 (0.068) | 0.736 (0.090) |
| | X | N | 0.516 (0.084) | 0.914 (0.069) | 0.515 (0.045) | 1.248 (0.085) | 0.516 (0.064) | 1.081 (0.077) |
| | | L | 0.498 (0.087) | 0.929 (0.078) | 0.514 (0.050) | 1.255 (0.081) | 0.506 (0.068) | 1.092 (0.080) |
| **Neural LSDE** | O | N | 0.604 (0.081) | 0.752 (0.084) | **0.783 (0.031)** | **0.572 (0.080)** | 0.694 (0.056) | 0.662 (0.082) |
| | | L | 0.533 (0.089) | 0.877 (0.086) | 0.745 (0.038) | 0.668 (0.085) | 0.639 (0.064) | 0.772 (0.085) |
| | X | N | 0.530 (0.069) | 0.856 (0.060) | 0.527 (0.054) | 1.177 (0.082) | 0.528 (0.060) | 1.039 (0.074) |
| | | L | 0.505 (0.064) | 0.912 (0.061) | 0.518 (0.057) | 1.210 (0.101) | 0.512 (0.059) | 1.083 (0.088) |
| **Neural LNSDE** | O | N | **0.654 (0.073)** | **0.701 (0.091)** | 0.780 (0.029) | 0.577 (0.070) | **0.717 (0.051)** | **0.639 (0.080)** |
| | | L | 0.586 (0.087) | 0.765 (0.083) | 0.764 (0.044) | 0.617 (0.108) | 0.675 (0.066) | 0.691 (0.096) |
| | X | N | 0.532 (0.077) | 0.878 (0.060) | 0.502 (0.051) | 1.254 (0.082) | 0.517 (0.064) | 1.066 (0.071) |
| | | L | 0.528 (0.085) | 0.890 (0.069) | 0.510 (0.047) | 1.257 (0.089) | 0.519 (0.066) | 1.074 (0.079) |
| **Neural GSDE** | O | N | 0.633 (0.091) | 0.742 (0.086) | 0.772 (0.036) | 0.598 (0.077) | 0.703 (0.063) | 0.670 (0.081) |
| | | L | 0.572 (0.083) | 0.796 (0.084) | 0.748 (0.039) | 0.653 (0.094) | 0.660 (0.061) | 0.724 (0.089) |
| | X | N | 0.531 (0.074) | 0.868 (0.052) | 0.510 (0.045) | 1.261 (0.093) | 0.520 (0.060) | 1.064 (0.072) |
| | | L | 0.525 (0.078) | 0.893 (0.072) | 0.509 (0.052) | 1.261 (0.093) | 0.517 (0.065) | 1.077 (0.083) |

## 5 CONCLUSION

In this study, we proposed three stable classes of Neural SDEs - Langevin-type SDE, Linear Noise SDE, and Geometric SDE - with the aim of capturing complex dynamics and improving stability in time series data. While the drift and diffusion terms in the existing Neural SDEs are directly approximated by neural networks, the proposed Neural SDEs are trained based on theoretically well-defined SDEs. We investigated the theoretical properties of the proposed Neural SDE classes, particularly their robustness under distribution shift, and corroborated their effectiveness in handling real-world irregular time series data through extensive experiments. As a result, the proposed Neural SDEs achieved state-of-the-art results in a wide range of experiments. However, it is important to acknowledge that our methods require more computational resources compared to Neural CDE-based models. Despite this, we found that our methods significantly enhance the stability of Neural SDE training and improve classification performance under challenging circumstances.

ETHIC STATEMENT

We commit to conducting our research with integrity, ensuring ethical practices and responsible use of technology in alignment with established academic and scientific standards.

REPRODUCIBILITY STATEMENT

We make our code and implementation details publicly accessible for reproducibility purposes. Code is available at `https://github.com/yongkyung-oh/Stable-Neural-SDEs`.

ACKNOWLEDGEMENT

We thank the teams and individuals for their efforts in dataset preparation and curation for our research, especially the UEA & UCR repository for the 30 datasets we extensively analyzed.

This work was partly supported by the Korea Health Technology R&D Project through the Korea Health Industry Development Institute (KHIDI), funded by the Ministry of Health and Welfare, Republic of Korea (Grant number: HI19C1095), the National Research Foundation of Korea (NRF) grant funded by the Korea government (MSIT)(No.RS-2023-00253002), the National Research Foundation of Korea (NRF) grant funded by the Korea government (MSIT)(No.RS-2023-00218913), and the Institute of Information & communications Technology Planning & Evaluation (IITP) grant funded by the Korea government (MSIT) (No.2020-0-01336, Artificial Intelligence Graduate School Program (UNIST)).

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

## A    PROOFS FOR SECTION 3.2

It is a classical result that SDE has a unique strong solution when the drift and diffusion functions are Lipschitz continuous and at most linearly growing as stated in the following Theorem.

**Theorem A.1.** *(Mao (2007)) Consider the following d-dimensional stochastic differential equation*

$$\mathrm{d}X(t) = f(t, X(t))\mathrm{d}t + g(t, X(t))\mathrm{d}W(t)$$

*where $W(t)$ is the d-dimensional Brownian motion. Assume that $f(t, x), g(t, x)$ are Lipschitz continuous in $(t, x)$ and satisfy the linear growth condition for every finite subinterval $[0, T]$, i.e., there exist constants $K_T, L_T > 0$ such that for all $t \in [0, T]$ and $x, x' \in \mathbb{R}^d$,*

$$|f(t, x) - f(t, x')| + |g(t, x) - (t, x')| \leq L_T|x - x'|,$$

*and*

$$|f(t, x)| + |g(t, x)| \leq K_T(1 + |x|).$$

*Then, there exists a unique strong (global) solution $\{X(t)\}_{t \geq 0}$ satisfying*

$$\mathbb{E}\left[\int_0^\infty |X(t)|^2 \mathrm{d}t\right] < \infty.$$

***Proof of Proposition 3.4.*** (i) For the Langevin SDE and linear noise SDE, from Assumptions 3.1 and 3.2, the drift and diffusion coefficients are Lipschitz continuous and at most linearly growing for every finite subinterval $t \in [0, T]$. Thus, they have their unique strong solutions by Theorem A.1.

(ii) Recall that we consider `tanh` or `sigmoid` functions as activation functions for the GSDE. Hence, the drift and diffusion coefficients in Equation (6) satisfy the conditions stated in Theorem A.1. Therefore, Equation (6) has a unique strong global solution.

To prove the nonnegativity of the solution, consider a stochastic process $\{\boldsymbol{y}(t)\}_{t \geq 0}$ defined on the same probability space, which is governed by the following SDE:

$$\mathrm{d}\boldsymbol{y}(t) = \left(\gamma(t, e^{\boldsymbol{y}(t)}; \theta_\gamma) - \frac{1}{2}\operatorname{diag}(\sigma(t; \theta_\sigma)\sigma(t; \theta_\sigma)^\top)\right)\mathrm{d}t + \sigma(t; \theta_\sigma)\mathrm{d}W(t). \qquad (11)$$

where $\operatorname{diag}(A)$ is the vector of the diagonal elements of matrix $A$. Since the diffusion term of Equation (11) is Lipschitz continuous and is at most linearly growing, from Theorem A.1, Equation (11) has a unique strong solution, so $\{\boldsymbol{y}(t)\}_{t \geq 0}$ is well-defined.

Let $\boldsymbol{z}(t) = \exp(\boldsymbol{y}(t))$, i.e. $\boldsymbol{z}_i(t) = \exp(\boldsymbol{y}_i(t))$ for $i = 1, 2, \ldots, d_z$ where $\boldsymbol{z}_i$ and $\boldsymbol{y}_i$ are the $i$-th components of $\boldsymbol{z}$ and $\boldsymbol{y}$, respectively. For all $i = 1, 2, \ldots, d_z$, we note that $\boldsymbol{z}_i(t)$ is nonnegative almost surely for all $t \geq 0$ by its construction. Also, using Ito's formula we have

$$\mathrm{d}\boldsymbol{z}_i(t) = e^{\boldsymbol{y}_i(t)}\mathrm{d}\boldsymbol{y}_i(t) + \frac{1}{2}e^{\boldsymbol{y}_i(t)}\mathrm{d}[\boldsymbol{y}_i(t), \boldsymbol{y}_i(t)]$$

$$= \boldsymbol{z}_i(t)\gamma_i(t, \boldsymbol{z}(t); \theta_f)\mathrm{d}t + \boldsymbol{z}_i(t)\sum_{j=1}^{d_z}(\sigma(t; \theta_\sigma))_{i,j}\mathrm{d}W_j(t).$$

where $\gamma_i$ is the $i$-th element of $\gamma$ and $(\sigma(t; \theta_\sigma))_{ij}$ represents the $(i, j)$-th element of $\sigma(t; \theta_\sigma)$, and $[\boldsymbol{y}_i(t), \boldsymbol{y}_i(t)]$ is the quadratic variation of $\boldsymbol{y}_i(t)$ up to $t$ for $i = 1, 2, \ldots, d_z$. In the matrix form, $\boldsymbol{z}(t)$ satisfies

$$\frac{\mathrm{d}\boldsymbol{z}(t)}{\boldsymbol{z}(t)} = \gamma(t, \boldsymbol{z}(t); \theta_f)\mathrm{d}t + \sigma(t; \theta_\sigma)\mathrm{d}W(t).$$

Therefore, $\boldsymbol{z}(t)$ is the solution of Equation (6). Let $T_0^i = \inf\{t > 0 | z_i(t) = 0\}$ be the first hitting time to state 0 with $z_i(0) > 0$ for $i = 1, 2, \ldots, d_z$ where $z_i(t)$ is the $i$-th component of $\boldsymbol{z}(t)$. Since $\mathrm{d}z_i(t) = 0$ for $t \geq T_0^i$, then $z_i(t) = 0$ for all $t \geq T_0^i$. This implies that once $z_i(t)$ reaches 0, it remains there forever. Therefore, 0 is an absorbing state. $\qquad \square$

## A.1 DETAILS TO REMARK 3.7

To incorporate a controlled path of the underlying time-series data, we replace $z(t)$ in the drift functions of the proposed Neural SDEs with $\overline{z}(t)$ defined in Equation (10). More specifically, when combined with $\overline{z}(t)$, the proposed Neural SDEs are given by

$$\text{(Neural LSDE)} \quad \mathrm{d}z(t) = \overline{\gamma}_1(z(t); \theta_\gamma, \theta_\zeta)\mathrm{d}t + \sigma(t; \theta_\sigma)\mathrm{d}W(t), \tag{12}$$

$$\text{(Neural LNSDE)} \quad \mathrm{d}z(t) = \overline{\gamma}_2(t, z(t); \theta_\gamma, \theta_\zeta)\mathrm{d}t + \sigma(t; \theta_\sigma)z(t)\mathrm{d}W(t), \tag{13}$$

$$\text{(Neural GSDE)} \quad \frac{\mathrm{d}z(t)}{z(t)} = \overline{\gamma}_2(t, z(t); \theta_\gamma, \theta_\zeta)\mathrm{d}t + \sigma(t; \theta_\sigma)\mathrm{d}W(t). \tag{14}$$

where $\overline{\gamma}_1(z(t); \theta_\gamma, \theta_\zeta) := \gamma(\overline{z}(t); \theta_\gamma) = \gamma(\zeta(t, z(t), X(t); \theta_\zeta); \theta_\gamma)$ and $\overline{\gamma}_2(t, z(t); \theta_\gamma, \theta_\zeta) := \gamma(t, \overline{z}(t); \theta_\gamma) = \gamma(t, \zeta(t, z(t), X(t); \theta_\zeta); \theta_\gamma)$ for given $\{X(t)\}_{t \geq 0}$, and $\zeta$ is a neural network with parameter $\theta_\zeta$.

We highlight that the proposed Neural SDEs defined in Equations 12, 13, and 14 have their unique strong solutions. To show this, it is enough to show that $\overline{\gamma}_1(\cdot; \theta_\gamma, \theta_\zeta)$ and $\overline{\gamma}_2(\cdot, \cdot; \theta_\gamma, \theta_\zeta)$ are Lipschitz continuous and at most linearly growing (or bounded for GSDE).

First of all, one can show that $\overline{\gamma}_1(\cdot; \theta_\gamma, \theta_\zeta)$ is Lipschitz continuous for $x, y \in \mathbb{R}^d$ since

$$
\begin{aligned}
|\overline{\gamma}_1(x; \theta_\gamma, \theta_\zeta) - \overline{\gamma}_1(y; \theta_\gamma, \theta_\zeta)| &\leq |\gamma(\zeta(t, x, X(t); \theta_\zeta); \theta_\gamma) - \gamma(\zeta(t, y, X(t); \theta_\zeta); \theta_\gamma)| \\
&\leq L_\gamma |\zeta(t, x, X(t); \theta_\zeta) - \zeta(t, y, X(t); \theta_\zeta)| \\
&\leq L_\gamma L_\zeta |x - y|,
\end{aligned}
$$

where $L_\gamma$ and $L_\zeta$ are Lipschitz constants for the neural networks $\gamma$ and $\zeta$, respectively. In addition, $\overline{\gamma}_1(\cdot; \theta_\gamma, \theta_\zeta)$ is at most linearly growing since for all $x \in \mathbb{R}^d$,

$$
\begin{aligned}
|\overline{\gamma}_1(x; \theta_\gamma, \theta_\zeta)| = |\gamma(\zeta(x; \theta_\zeta); \theta_\gamma)| &\\
&\leq K_\gamma(1 + |\zeta(x; \theta_\zeta)|) \\
&= K_\gamma + K_\gamma K_\zeta(1 + |x|) \\
&\leq K_\gamma(1 + K_\zeta)(1 + |x|).
\end{aligned}
$$

Therefore, due to Theorem A.1, Neural LSDE combined with $\overline{z}(t)$ defined in Equation (12) has a unique strong solution. The proofs for Neural LNSDE and Neural can be shown in the same manner.

## B STOCHASTIC STABILITY AND PROOFS FOR SECTION 3.3

### B.1 BACKGROUND ON STABILITY OF SDEs

Stability is an important concept that describes the behavior of a solution in a given differential equation with respect to changes in the initial conditions. It is well-known that stability of ODEs can be determined without solving the equation by using the Lyapunov direct method. Fortunately, it turns out that the Lyapunov technique for analyzing stability of ODEs can be applied to study stability of SDEs with slight modifications. This is significantly useful since explicit solutions for SDEs cannot be obtained except in special cases. In this Appendix, we provide a brief overview of stochastic stability. We refer to Mao (2007); Khasminskii (2011) for more details.

Consider the $d$-dimensional SDE:

$$\mathrm{d}X(t) = f(t, X(t))\mathrm{d}t + g(t, X(t))\mathrm{d}W(t), \tag{15}$$

with the initial value $x_0 \in \mathbb{R}^d$. We assume that Equation (15) has a unique global solution denoted by $X(t; x_0)$ for all $t \geq 0$ with the initial value $x_0 = 0$. Furthermore, assume that

$$f(t, 0) = 0 \quad \text{and} \quad g(t, 0) = 0$$

for $t \geq 0$. Then, $X(t; x_0) \equiv 0$ is the solution of Equation (15) and is called the trivial solution. For $V \in C^{1,2}(\mathbb{R}_+ \times \mathbb{R}^d)$, we define the infinitesimal generator $\mathcal{G}$ of $z(t)$:

$$\mathcal{G}V(t, x) = V_t(t, x) + V_x(t, x)^\top f(t, x) + \frac{1}{2}\mathrm{Tr}[g(t, x)g(t, x)^\top V_{xx}(t, x)]$$

where $V_x$ is the gradient of $V$ with respect to $x$ and $V_{xx}$ is the Hessian of $V$ with respect to $x$.

In a stochastic system driven by a SDE, stability can be defined in several ways such as stability in probability, almost sure exponential stability, and moment exponential stability. For our purpose, we focus on almost sure exponential stability.

**Definition B.1.** The trivial solution of Equation (15) is *almost surely exponential stable* if for all $x_0 \in \mathbb{R}^d$

$$\limsup_{t \to \infty} \frac{1}{t} \log |X(t; x_0)| < 0 \quad a.s.$$

The almost surely exponential stability implies that almost all sample paths of the solution will converge to $x = 0$ exponentially fast. We record a criterion (sufficient condition) for the almost surely exponential stability.

**Theorem B.2.** *(Mao (2007)) Assume that there is a function $V(t, x) \in \mathcal{C}^{1,2}(\mathbb{R}_+ \times \mathbb{R}^d)$, and constants $p > 0, c_1, c_2 \in \mathbb{R}, c_3 \geq 0$ such that*

    *(i)* $c_1 |x|^p \leq V(t, x)$,

    *(ii)* $\mathcal{G}V(t, x) \leq c_2 V(t, x)$,

    *(iii)* $|V_x(t, x) g(t, x)|^2 \geq c_3 V^2(t, x)$,

*for all $(t, x) \in \mathbb{R}_+ \times \mathbb{R}^d$. Then,*

$$\limsup_{t \to \infty} \frac{1}{t} \log |X(t; x_0)| \leq -\frac{c_3 - 2c_2}{2p} \quad a.s.$$

*for all $x_0 \in \mathbb{R}^d$. In particular, if $c_3 > 2c_2$, the given SDE is almost surely exponentially stable.*

A function $V(t, x)$ that satisfies the stability conditions of Theorem B.2 is called a Lyapunov function. Theorem B.2 states that one can determine stability of given SDEs by specifying a suitable Lyapunov function without solving the solution explicitly.

## B.2 LANGEVIN SDE.

The Langevin SDE has been extensively studied in the field of stochastic optimization and MCMC algorithms because it admits a unique invariant measure, which is indeed a Gibbs measure. For more detailed discussions, we refer to Raginsky et al. (2017); Chau et al. (2021); Lim & Sabanis (2021); Lim et al. (2023a;b).

Recall that we consider the following SDE:

$$d\boldsymbol{z}(t) = \gamma(\boldsymbol{z}(t)) dt + \sigma_\theta dW(t), \tag{16}$$

where $\sigma_\theta \in \mathbb{R}$, which admits the unique invariant measure $\pi$.

For $p \geq 1$, we define a Lyapunov function $V_p$ by

$$V_p(x) := (1 + |x|^2)^{p/2}, \tag{17}$$

for $x \in \mathbb{R}^{d_z}$. Let $\mathcal{P}_{V_p}(\mathbb{R}^d)$ be the set of $\mu \in \mathcal{P}(\mathbb{R}^d)$ satisfying $\int_{\mathbb{R}^d} V_p(x)\mu(dx)$. We then introduce a functional which shows geometric ergodicity of the Langevin SDE. For $p \geq 1$, $\mu, \nu \in \mathcal{P}_{V_p}$, define

$$w_{1,p}(\mu, \nu) := \inf_{\Pi \in \mathcal{C}(\mu,\nu)} \left( \int_{\mathbb{R}^d} \int_{\mathbb{R}^d} [1 \wedge |x - x'|] \left(1 + V_p(x) + V_p(x')\right) \Pi(dx, dx') \right).$$

Notice that one can easily show that $\mathcal{W}_1(\mu, \nu) \leq w_{1,p}(\mu, \nu)$. The key to analyzing stability of the Neural SDE based on the Langevin SDE is a contraction property of $w_{1,2}$.

**Proposition B.3.** *(Chau et al. (2021)) Let Assumptions 3.1, 3.2 and 3.3 hold. Let $\boldsymbol{z}(t), \boldsymbol{z}'(t)$ be the solutions of Equation (16) with the initial values $z(0), z'(0) \in L^4$, respectively. Then, there exists positive constants $c_1, c_2 > 0$ such that*

$$w_{1,2}(\mathcal{L}(\boldsymbol{z}(t), \mathcal{L}(\boldsymbol{z}'(t))) \leq c_1 e^{-c_2 t} w_{1,2}(\mathcal{L}(\boldsymbol{z}(0))), \mathcal{L}(\boldsymbol{z}'(0)))$$

*for all $t > 0$.*

Now we derive our main result using the contraction property stated in Proposition B.3.

**_Proof of Theorem 3.5_.** Fix $T > 0$. Recall that $\boldsymbol{y}$, $\widetilde{\boldsymbol{y}}$ represent the outputs of the Neural LSDE with the inputs $\boldsymbol{x}$, $\widetilde{\boldsymbol{x}}$, respectively. Assume that $\sqrt{\mathbb{E}[|\boldsymbol{x} - \widetilde{\boldsymbol{x}}|^2]} \leq \rho$. For notational convenience, we drop the dependence on $\theta_F$ in Equation (9). Due to Assumption 3.1, there exists $L_F > 0$ such that for all $x, x' \in \mathbb{R}^{d_z}$

$$|F(x) - F(x')| \leq L_F |x - x'|.$$

Then, we observe that

$$\mathcal{W}_1(\mathcal{L}(\boldsymbol{y}), \mathcal{L}(\widetilde{\boldsymbol{y}})) = \mathcal{W}_1\left(\mathcal{L}(F(\boldsymbol{z}(T))), \mathcal{L}(F(\widetilde{\boldsymbol{z}}(T)))\right)$$

$$= \inf_{\Pi \in \mathcal{C}} \left( \int_{\mathbb{R}^d} \int_{\mathbb{R}^d} |F(\boldsymbol{z}) - F(\widetilde{\boldsymbol{z}})| \Pi(\mathrm{d}\boldsymbol{z}, \mathrm{d}\widetilde{\boldsymbol{z}}) \right)$$

$$\leq L_F \mathcal{W}_1\left(\mathcal{L}(\boldsymbol{z}(T)), \mathcal{L}(\widetilde{\boldsymbol{z}}(T))\right)$$

where $\mathcal{C}$ is the set of probability measures such that its respective marginals are $\mathcal{L}(\boldsymbol{z}(T))$ and $\mathcal{L}(\widetilde{\boldsymbol{z}}(T))$.

Furthermore, one can obtain the bound for $\mathcal{W}_1\left(\mathcal{L}(\boldsymbol{z}(T)), \mathcal{L}(\widetilde{\boldsymbol{z}}(T))\right)$ using Proposition B.3, Cauchy-Schwartz inequality, Jensen's inequality and the definition of $V_2(\cdot)$

$$\mathcal{W}_1\left(\mathcal{L}(\boldsymbol{z}(T)), \mathcal{L}(\widetilde{\boldsymbol{z}}(T))\right) \leq w_{1,2}\left(\mathcal{L}(\boldsymbol{z}(T)), \mathcal{L}(\widetilde{\boldsymbol{z}}(T))\right)$$

$$\leq c_1 e^{-c_2 T} w_{1,2}\left(\mathcal{L}(\boldsymbol{z}(0))), \mathcal{L}(\widetilde{\boldsymbol{z}}(0))\right)$$

$$\leq L_h c_1 e^{-c_2 T} w_{1,2}(\mathcal{L}(\boldsymbol{x}), \mathcal{L}(\widetilde{\boldsymbol{x}}))$$

$$\leq L_h c_1 e^{-c_2 T} \mathbb{E}\left[(1 \wedge |\boldsymbol{x} - \widetilde{\boldsymbol{x}}|)\left(1 + V_2(\boldsymbol{x}) + V_2(\widetilde{\boldsymbol{x}})\right)\right]$$

$$\leq L_h c_1 e^{-c_2 T} \sqrt{\mathbb{E}[|\boldsymbol{x} - \widetilde{\boldsymbol{x}}|^2]} \sqrt{\mathbb{E}[(1 + V_2(\boldsymbol{x}) + V_2(\widetilde{\boldsymbol{x}}))^2]}$$

$$\leq \sqrt{3} L_h c_1 e^{-c_2 T} \sqrt{\mathbb{E}[|\boldsymbol{x} - \widetilde{\boldsymbol{x}}|^2]} \sqrt{\mathbb{E}[(1 + V_2^2(\boldsymbol{x}) + V_2^2(\widetilde{\boldsymbol{x}}))]}$$

$$\leq \sqrt{3} L_h c_1 e^{-c_2 T} \rho \sqrt{1 + \mathbb{E}[V_2^2(\boldsymbol{x})] + \mathbb{E}[V_2^2(\widetilde{\boldsymbol{x}})]}$$

$$\leq \sqrt{3} L_h c_1 e^{-c_2 T} \sqrt{(5 + 2\mathbb{E}[|\boldsymbol{x}|^4] + 2\mathbb{E}[|\widetilde{\boldsymbol{x}}|^4])} \rho$$

where $L_h$ is the Lipschitz constant of the initial condition $h$ defined in Equation (4).

Combining the two inequalities above, it follows that

$$\mathcal{W}_1(\mathcal{L}(\boldsymbol{y}), \mathcal{L}(\widetilde{\boldsymbol{y}})) \leq \sqrt{3} L_F L_h c_1 e^{-c_2 T} \sqrt{(5 + 2\mathbb{E}[|\boldsymbol{x}|^2] + 2\mathbb{E}[|\widetilde{\boldsymbol{x}}|^2])} \rho.$$

Now focus on deriving the upper bound for $\mathcal{W}_2(\mathcal{L}(\boldsymbol{y}), \mathcal{L}(\widetilde{\boldsymbol{y}}))$. Observe that

$$\mathcal{W}_2^2(\mathcal{L}(\boldsymbol{y}), \mathcal{L}(\widetilde{\boldsymbol{y}})) = \mathcal{W}_2^2\left(\mathcal{L}(F(\boldsymbol{z}(T))), \mathcal{L}(F(\widetilde{\boldsymbol{z}}(T)))\right)$$

$$= \inf_{\Pi \in \mathcal{C}} \left( \int_{\mathbb{R}^d} \int_{\mathbb{R}^d} |F(\boldsymbol{z}) - F(\widetilde{\boldsymbol{z}})|^2 \Pi(\mathrm{d}\boldsymbol{z}, \mathrm{d}\widetilde{\boldsymbol{z}}) \right)$$

$$\leq L_F^2 \mathcal{W}_2^2\left(\mathcal{L}(\boldsymbol{z}(T)), \mathcal{L}(\widetilde{\boldsymbol{z}}(T))\right). \tag{18}$$

From the definition of the Wassersten-2 distance, we write for $\mu, \nu \in \mathcal{P}_{V_2}(\mathbb{R}^d)$

$$\mathcal{W}_2(\mu, \nu)^2 \leq \int_{\mathbb{R}^d} \int_{\mathbb{R}^d} |x - x'|^2 \Pi(\mathrm{d}x, \mathrm{d}x')$$

$$\leq \int_{\mathbb{R}^d} \int_{\mathbb{R}^d} |x - x'|^2 \mathbf{1}_{|x-x'| \geq 1} \Pi(\mathrm{d}x, \mathrm{d}x') + \int_{\mathbb{R}^d} \int_{\mathbb{R}^d} |x - x'|^2 \mathbf{1}_{|x-x'| < 1} \Pi(\mathrm{d}x, \mathrm{d}x')$$

$$\leq 2 \int_{\mathbb{R}^d} \int_{\mathbb{R}^d} (|x|^2 + |x'|^2) \mathbf{1}_{|x-x'| \geq 1} \Pi(\mathrm{d}x, \mathrm{d}x') + 2 \int_{\mathbb{R}^d} \int_{\mathbb{R}^d} |x - x'|(|x| + |x'|) \mathbf{1}_{|x-x'| < 1} \Pi(\mathrm{d}x, \mathrm{d}x')$$

$$\leq 2 \int_{\mathbb{R}^d} \int_{\mathbb{R}^d} [1 \wedge |x - x'|](1 + V_2(x) + V_2(x')) \Pi(\mathrm{d}x, \mathrm{d}x').$$

Taking infimum over $\Pi \in \mathcal{C}(\mu, \nu)$, we have

$$\mathcal{W}_2(\mu, \nu) \leq \sqrt{2 w_{1,2}(\mu, \nu)}.$$

Therefore, one further calculates that from Proposition B.3, Cauchy-Schwartz inequality and Jensen's inequality

$$
\begin{aligned}
\mathcal{W}_2^2\left(\mathcal{L}(\boldsymbol{z}(T)),\mathcal{L}(\widetilde{\boldsymbol{z}}(T))\right) &\leq 2w_{1,2}\left(\mathcal{L}(\boldsymbol{z}(T)),\mathcal{L}(\widetilde{\boldsymbol{z}}(T))\right) \\
&\leq 2c_1 e^{-c_2 T} w_{1,2}\left(\mathcal{L}(\boldsymbol{z}(0))),\mathcal{L}(\widetilde{\boldsymbol{z}}(0))\right) \\
&\leq 2L_h c_1 e^{-c_2 T} w_{1,2}(\mathcal{L}(\boldsymbol{x}),\mathcal{L}(\widetilde{\boldsymbol{x}})) \\
&\leq 2L_h c_1 e^{-c_2 T}\mathbb{E}\left[(1\wedge|\boldsymbol{x}-\widetilde{\boldsymbol{x}}|)\left(1+V_2(\boldsymbol{x})+V_2(\widetilde{\boldsymbol{x}})\right)\right] \\
&\leq 2\sqrt{3}L_h c_1 e^{-c_2 T}\sqrt{\mathbb{E}[|\boldsymbol{x}-\widetilde{\boldsymbol{x}}|^2]}\sqrt{\mathbb{E}[(1+V_2^2(\boldsymbol{x})+V_2^2(\widetilde{\boldsymbol{x}}))]} \\
&\leq 2\sqrt{3}L_h c_1 e^{-c_2 T}\sqrt{(5+2\mathbb{E}[|\boldsymbol{x}|^4]+2\mathbb{E}[|\widetilde{\boldsymbol{x}}|^4])}\rho
\end{aligned}
\tag{19}
$$

From Equation (18) and Equation (19), we derive

$$
\mathcal{W}_2^2(\mathcal{L}(\boldsymbol{y}),\mathcal{L}(\widetilde{\boldsymbol{y}})) \leq 2\sqrt{3}L_F^2 L_h c_1 e^{-c_2 T}\sqrt{(5+2\mathbb{E}[|\boldsymbol{x}|^4]+2\mathbb{E}[|\widetilde{\boldsymbol{x}}|^4])}\rho.
$$

$\square$

### B.3 LINEAR NOISE SDE AND GEOMETRIC SDE

**Lemma B.4.** *Consider the following $d$-dimensional SDE:*

$$
\mathrm{d}X(t) = f(t, X(t))\mathrm{d}t + \sigma_\theta X(t)\mathrm{d}W(t)
$$

*with the initial condition $X(0) \in L^2$. Assume that $f$ satisfies the conditions in Assumptions 3.1 and 3.2. Furthermore, the drift function $f$ satisfies the condition in Assumption 3.3. That is, there exists positive constants $m > 0$ and $b \geq 0$ such that*

$$
\langle f(t,x),\, x \rangle \leq -m|x|^2 + b.
\tag{20}
$$

*for all $x \in \mathbb{R}^d$. Then,*

$$
\sup_{t\geq 0}\mathbb{E}[|X(t)|^2] \leq \left(\mathbb{E}[|X(0)|^2] + \frac{b}{m}\right)\exp\left\{\frac{d\sigma_\theta^2}{2m}\right\}.
$$

*and*

$$
\sup_{t\geq 0}\mathbb{E}[|X(t)|] \leq \sqrt{\left(\mathbb{E}[|X(0)|^2] + \frac{b}{m}\right)\exp\left\{\frac{d\sigma_\theta^2}{2m}\right\}}.
$$

***Proof of Lemma B.4.*** Let $Y(t) := |X(t)|^2$ for all $t \geq 0$. Then, Itô's formula gives for

$$
\begin{aligned}
\mathrm{d}\left(e^{2mt}Y(t)\right) &= 2me^{2mt}Y(t)\mathrm{d}t + 2e^{2mt}\langle f(t,X(t)),\, X(t)\rangle\mathrm{d}t \\
&\quad + e^{2mt}d\sigma_\theta^2\mathrm{Tr}(X(t)X(t)^\top)\mathrm{d}t + 2\langle \sigma_\theta X(t),\, X(t)\rangle\mathrm{d}W_t,
\end{aligned}
\tag{21}
$$

which yields

$$
\begin{aligned}
Y(t) &= e^{-2mt}Y(0) + 2m\int_0^t e^{2m(s-t)}Y(s)\mathrm{d}s + 2\int_0^t e^{2m(s-t)}\langle f(s,X(s)),\, X(s)\rangle\mathrm{d}s \\
&\quad + d\sigma_\theta^2\int_0^t e^{2m(s-t)}Y(s)\mathrm{d}s + 2\sigma_\theta e^{-2mt}\int_0^t Y(s)\mathrm{d}W_s.
\end{aligned}
\tag{22}
$$

Due to Equation (20), we have

$$
\begin{aligned}
\int_0^t e^{2m(s-t)}\langle f(s,X(s)),\, X(s)\rangle\mathrm{d}s &\leq -m\int_0^t e^{2m(s-t)}|X(s)|^2\mathrm{d}s + b\int_0^t e^{2m(s-t)}\mathrm{d}s \\
&= -m\int_0^t e^{2m(s-t)}Y(s)\mathrm{d}s + \frac{b}{2m}(1-e^{-2mt}).
\end{aligned}
\tag{23}
$$

Substituting Equation (23) into Equation (22), we have

$$Y(t) \leq e^{-2mt} Y(0) + \frac{b}{m}(1 - e^{-2mt})$$
$$+ d\sigma_\theta^2 \int_0^t e^{2m(s-t)} Y(s) \mathrm{d}s + 2\sigma_\theta e^{-2mt} \int_0^t Y(s) \mathrm{d}W_s,$$

and then taking expectations yields

$$\mathbb{E}[Y(t)] \leq e^{-2mt} \mathbb{E}[Y(0)] + \frac{b}{m}(1 - e^{-2mt}) + d\sigma_\theta^2 \int_0^t e^{2m(s-t)} \mathbb{E}[Y(s)] \mathrm{d}s.$$

Using Gronwall's inequality leads to

$$\mathbb{E}[Y(t)] \leq \left( \mathbb{E}[Y(0)] + \frac{b}{m}(1 - e^{-2mt}) \right) \exp\left\{ d\sigma_\theta^2 \int_0^t e^{2m(s-t)} \mathrm{d}s \right\}$$
$$\leq \left( \mathbb{E}[Y(0)] + \frac{b}{m}(1 - e^{-2mt}) \right) \exp\left\{ \frac{d\sigma_\theta^2}{2m}(1 - e^{-2mt}) \right\}$$
$$\leq \left( \mathbb{E}[Y(0)] + \frac{b}{m} \right) \exp\left\{ \frac{d\sigma_\theta^2}{2m} \right\}.$$

Lastly, we have

$$\mathbb{E}[|X(t)|] \leq \sqrt{\mathbb{E}[|X(t)|^2]} \leq \sqrt{\left( \mathbb{E}[Y(0)] + \frac{b}{m} \right) \exp\left\{ \frac{d\sigma_\theta^2}{2m} \right\}}.$$

$\square$

***Proof of Theorem 3.6.*** (i) **Linear Noise SDE.** Using the same argument in the Proof of Theorem 3.5, we have

$$\mathcal{W}_1(\mathcal{L}(\boldsymbol{y}), \mathcal{L}(\widetilde{\boldsymbol{y}})) = \mathcal{W}_1\left(\mathcal{L}(F(\boldsymbol{z}(T))), \mathcal{L}(F(\widetilde{\boldsymbol{z}}(T)))\right)$$
$$= \inf_{\Pi \in \mathcal{C}} \left( \int_{\mathbb{R}^d} \int_{\mathbb{R}^d} |F(\boldsymbol{z}) - F(\widetilde{\boldsymbol{z}})| \Pi(\mathrm{d}\boldsymbol{z}, \mathrm{d}\widetilde{\boldsymbol{z}}) \right) \tag{24}$$
$$\leq L_F \mathcal{W}_1\left(\mathcal{L}(\boldsymbol{z}(T)), \mathcal{L}(\widetilde{\boldsymbol{z}}(T))\right) \tag{25}$$

where $\mathcal{C}$ is the set of probability measures such that its respective marginals are $\mathcal{L}(\boldsymbol{z}(T))$ and $\mathcal{L}(\widetilde{\boldsymbol{z}}(T))$.

Now, we show that the solution of Equation (5) is almost surely moment exponential stable defined in Definition B.1. Let $\boldsymbol{z}(t), \widetilde{\boldsymbol{z}}(t)$ be the solutions of Equation (5) with the initial conditions $\boldsymbol{z}(0), \widetilde{\boldsymbol{z}}(0)$, respectively. Define $\varepsilon(t) = \boldsymbol{z}(t) - \widetilde{\boldsymbol{z}}(t)$ for all $t \geq 0$. Then, $\varepsilon(t)$ is the solution of the following SDE:

$$\mathrm{d}\varepsilon(t) = (\gamma(t, \boldsymbol{z}(t); \theta_\gamma) - \gamma(t, \widetilde{\boldsymbol{z}}(t); \theta_\gamma)) \mathrm{d}t + \sigma_\theta (\boldsymbol{z}(t) - \widetilde{\boldsymbol{z}}(t)) \mathrm{d}W(t)$$
$$= \gamma_\Delta(t, \varepsilon(t)) \mathrm{d}t + \sigma_\Delta(t, \varepsilon(t)) \mathrm{d}W_t, \tag{26}$$

where

$$\gamma_\Delta(t, \varepsilon(t)) := \gamma(t, \widetilde{\boldsymbol{z}}(t) + \varepsilon(t); \theta_\gamma) - \gamma(t, \widetilde{\boldsymbol{z}}(t); \theta_\gamma) = \gamma(t, \boldsymbol{z}(t); \theta_\gamma) - \gamma(t, \widetilde{\boldsymbol{z}}(t); \theta_\gamma),$$

and

$$\sigma_\Delta(t, \varepsilon(t)) := \sigma_\theta \varepsilon(t).$$

Note that $\gamma_\Delta(t, 0) = 0$ and $\sigma_\Delta(t, 0) = 0$ for all $t$. Hence, Equation (26) has the trivial solution $\varepsilon(t) = 0$ when $\varepsilon(0) = 0$ is given. Then, due to Assumptions 3.1 and 3.2, one can show that for all $x, x' \in \mathbb{R}^d$

$$|\gamma_\Delta(t, x) - \gamma_\Delta(t, x')| \leq L_\gamma |x - x'|,$$
$$|\sigma_\Delta(t, x) - \sigma_\Delta(t, x')| \leq \sigma_\theta |x - x'|$$
$$|\gamma_\Delta(t, x)| \leq L_\gamma |x|,$$
$$|\sigma_\Delta(t, x)| = \sigma_\theta |x|,$$

yielding that the Equation (26) has a unique strong solution from Theorem A.1.

We introduce a Lyapunov function $V(t, x) = |x|^2$. Then, we have

$$\mathcal{G}V(t, x) = V_x(t, x)^\top \gamma_\Delta(t, x) + \frac{1}{2} \text{Tr}[\sigma_\Delta(t, x)\sigma_\Delta(t, x)^\top V_{xx}(t, x)]$$

$$\leq (2L_\gamma + |\sigma_\theta|^2)|x|^2 = (2L_\gamma + |\sigma_\theta|^2)V(t, x),$$

$$|V_x(t, x)\sigma_\Delta(t, x)|^2 = 4|\sigma_\theta|^2|x|^4 = 4|\sigma_\theta|^2 V(t, x)^2.$$

Using Theorem B.2, we obtain

$$\limsup_{t \to \infty} \frac{1}{t} \log |\varepsilon(t; \varepsilon(0))| \leq -\frac{|\sigma_\theta|^2 - 2L_\gamma}{2} \quad a.s.$$

In other words, for $|\sigma_\theta|^2 > 2L_\gamma$, Equation (26) is almost surely exponential stable. Observe that

$$\sup_{t \geq 0} \mathbb{E}[|\varepsilon(t; \varepsilon(0))|] \leq \sup_{t \geq 0} \mathbb{E}[(|\boldsymbol{z}(t)| + |\widetilde{\boldsymbol{z}}(t)|)] < \infty,$$

due to Lemma B.4. Thus, by Egorov's theorem, there exists $T^* > 0$ and $E \in \mathcal{F}$ such that

$$\frac{1}{T} \log |\varepsilon(T; \varepsilon(0))| \leq -\frac{|\sigma_\theta|^2 - 2L_\gamma}{2}, \quad \omega \in E,$$

with $P(E) \geq 1 - \delta$, and

$$\frac{1}{T} \log |\varepsilon(T; \varepsilon(0))| > -\frac{|\sigma_\theta|^2 - 2L_\gamma}{2}, \quad \omega \in E^c,$$

with $P(E^c) \leq \delta$ for $T \geq T^*$. Therefore, we can write

$$\mathbb{E}[|\varepsilon(T; \varepsilon(0))|] = \mathbb{E}[|\varepsilon(T; \varepsilon(0))\mathbf{1}_E] + \mathbb{E}[|\varepsilon(T; \varepsilon(0))|\mathbf{1}_{E^c}]$$

$$\leq \exp\left\{-\frac{|\sigma_\theta|^2 - 2L_\gamma}{2}\right\} + \sqrt{\mathbb{E}[|\varepsilon(T; \varepsilon(0))]\mathbb{E}[\mathbf{1}_{E^c}]}$$

$$\leq \exp\left\{-\frac{|\sigma_\theta|^2 - 2L_\gamma}{2}\right\} + \sqrt{\sup_{t \geq 0} \mathbb{E}[|\varepsilon(t; \varepsilon(0))]}\sqrt{\delta} \tag{27}$$

Hence, for any $\epsilon > 0$, we can choose sufficiently large $T$ satisfying

$$\mathbb{E}[|\varepsilon(T; \varepsilon(0))|] \leq \exp\{-(|\sigma_\theta|^2 - 2L_\gamma)T/2\} + \epsilon, \tag{28}$$

which leads to

$$\mathbb{E}[|\varepsilon(T; \varepsilon(0))|] \leq \exp\{-(|\sigma_\theta|^2 - 2L_\gamma)T/2\}(1 + \rho), \tag{29}$$

Combining Equation (25) and Equation (29), we have the desired result

$$\mathcal{W}_1(\mathcal{L}(\boldsymbol{y}), \mathcal{L}(\widetilde{\boldsymbol{y}})) \leq L_F \exp\{-(|\sigma_\theta|^2 - 2L_\gamma)T/2\}(1 + \rho),$$

for sufficiently large $T$.

(ii) **Geometric SDE.** We take the same argument as in the case of Linear Noise SDEs and adopt the same notations. The only difference is that the drift term $\gamma_\Delta(t, \varepsilon(t))$ for $\varepsilon(t)$ is given by

$$\gamma_\Delta(t, \varepsilon(t)) := \gamma(t, \widetilde{\boldsymbol{z}}(t) + \varepsilon(t); \theta_\gamma)(\widetilde{\boldsymbol{z}}(t) + \varepsilon(t)) - \gamma(t, \widetilde{\boldsymbol{z}}(t); \theta_\gamma)\widetilde{\boldsymbol{z}}(t)$$

satisfying that

$$|\gamma_\Delta(t, x) - \gamma_\Delta(t, x')| \leq K_\gamma|x - x'|.$$

due to the use of the bounded activation functions. The remaining part can be proved in the same way of the case of Linear Noise SDEs. $\qquad \square$

## C    DESCRIPTION OF DATASETS

**PhysioNet Mortality.**    The 2012 PhysioNet Mortality dataset (Silva et al., 2012) is a compilation of multivariate time series data, encompassing 37 distinct variables derived from Intensive Care Unit (ICU) records. Each dataset entry comprises irregular and infrequent measurements taken during the initial 48-hour period post ICU admission. Adhering to the methodologies proposed by Rubanova et al. (2019), we've adjusted the observation timestamps, rounding them to the closest minute, resulting in a potential 2880 distinct measurement instances for each time series.

For our interpolation experiments, we utilized all 8000 instances with the experiment pipeline, which is suggested by Shukla & Marlin (2021). We base our predictions on a selected subset of data points and aim to determine values for the remaining time intervals. This interpolation is executed with an observation range that spans from $50\%$ to $90\%$ of the total data points. During testing, the models are conditioned on the observed values and tasked with deducing the values for the remaining time intervals within the test set. The efficacy of the models is evaluated using the Mean Squared Error (MSE) with five random initializations of model parameters.

**MuJoCo.**    The MuJoCo dataset, which stands for Multi-Joint dynamics with Contact (Todorov et al., 2012), employed in this study utilizes the Hopper model from the DeepMind control suite, as detailed in Tassa et al. (2018). The dataset comprises 10,000 simulations of the Hopper model, each forming a 14-dimensional time series with 100 regularly sampled data points.

In our experimental setup, the standard training and testing horizon involves analyzing the first 50 observations in a sequence to forecast the subsequent 10 observations. To simulate more complex scenarios, we introduce variability by randomly omitting $30\%$, $50\%$, and $70\%$ of the values in each sequence, as Jhin et al. (2023b). This approach creates a range of challenging environments, particularly focusing on irregular time-series forecasting. The MSE metric is utilized for evaluation.

**PhysioNet Sepsis.**    The 2019 PhysioNet / Computing in Cardiology (CinC) challenge on Sepsis prediction serves as the foundation for time-series classification experiments (Reyna et al., 2019). Sepsis, a life-threatening condition triggered by bacteria or toxins in the blood, is responsible for a significant number of deaths in the United States of America. The dataset used in these experiments contains 40,335 cases of patients admitted to intensive care units, with 34 time-dependent variables such as heart rate, oxygen saturation, and body temperature. The primary objective is to classify whether each patient has sepsis or not according to the sepsis-3 definition.

The PhysioNet dataset is an irregular time series dataset, as only $10\%$ of the values are sampled with their respective timestamps for each patient. To address this irregularity, two types of time-series classification are performed: (i) classification using observation intensity (OI), and (ii) classification without using observation intensity (no OI). Observation intensity is a measure of the degree of illness, and when incorporated, an index number is appended to each value in the time series. Due to the imbalanced nature of the data, the Area Under the Receiver Operating Characteristic curve (AUROC) score is employed to evaluate the performance.

**Speech Commands.**    The Speech Commands dataset is an extensive collection of one-second audio recordings that encompass spoken words and background noise (Warden, 2018). This dataset is comprised of 34,975 time-series samples, representing 35 distinct spoken words. To create a balanced classification problem, ten labels (including 'yes', 'no', 'up', 'down', 'left', 'right', 'on', 'off', 'stop', and 'go') were selected from the dataset. The dataset is preprocessed by calculating Mel-frequency cepstral coefficients, which are used as features to better represent the characteristics of the audio recordings and improve the performance of machine learning algorithms applied to the dataset. Each sample in the dataset has a time-series length of 161 and an input size of 20 dimensions.

**Robustness to missing data.**    We examined the performance of the proposed Neural SDEs on 30 datasets from the University of East Anglia (UEA) and the University of California Riverside (UCR) Time Series Classification Repository [7] (Bagnall et al., 2018) using the python library `sktime` (Löning et al., 2019). The archive was comprised of univariate and multivariate time series datasets from various real-world applications.

---

[7]http://www.timeseriesclassification.com/

Table 6: Data description for 'Robustness to missing data' experiments

| Dataset | Total number of samples | Number of classes | Dimension of time series | Length of time series |
|---|---|---|---|---|
| **ArrowHead** | 211 | 3 | 1 | 251 |
| **Car** | 120 | 4 | 1 | 577 |
| **Coffee** | 56 | 2 | 1 | 286 |
| **GunPoint** | 200 | 2 | 1 | 150 |
| **Herring** | 128 | 2 | 1 | 512 |
| **Lightning2** | 121 | 2 | 1 | 637 |
| **Lightning7** | 143 | 7 | 1 | 319 |
| **Meat** | 120 | 3 | 1 | 448 |
| **OliveOil** | 60 | 4 | 1 | 570 |
| **Rock** | 70 | 4 | 1 | 2844 |
| **SmoothSubspace** | 300 | 3 | 1 | 15 |
| **ToeSegmentation1** | 268 | 2 | 1 | 277 |
| **ToeSegmentation2** | 166 | 2 | 1 | 343 |
| **Trace** | 200 | 4 | 1 | 275 |
| **Wine** | 111 | 2 | 1 | 234 |
| **ArticularyWordRecognition** | 575 | 25 | 9 | 144 |
| **BasicMotions** | 80 | 4 | 6 | 100 |
| **CharacterTrajectories** | 2858 | 20 | 3 | 60-180 |
| **Cricket** | 180 | 12 | 6 | 1197 |
| **Epilepsy** | 275 | 4 | 3 | 206 |
| **ERing** | 300 | 6 | 4 | 65 |
| **EthanolConcentration** | 524 | 4 | 3 | 1751 |
| **EyesOpenShut** | 98 | 2 | 14 | 128 |
| **FingerMovements** | 416 | 2 | 28 | 50 |
| **Handwriting** | 1000 | 26 | 3 | 152 |
| **JapaneseVowels** | 640 | 9 | 12 | 7-26 |
| **Libras** | 360 | 15 | 2 | 45 |
| **NATOPS** | 360 | 6 | 24 | 51 |
| **RacketSports** | 303 | 4 | 6 | 30 |
| **SpokenArabicDigits** | 8798 | 10 | 13 | 4-93 |

According to Table 6, the datasets have distinct sample sizes, dimensions, lengths, and the number of classes. Some datasets contain variable-length samples. To tackle the issue of varying time series lengths, we applied uniform scaling (Keogh, 2003; Yankov et al., 2007; Gao & Lin, 2018; Tan et al., 2019) to match all series to the length of the longest one. Subsequently, we generated random missing observations for each time series and then combined the modified series. The data was divided into train, validation, and test sets in a 0.70/0.15/0.15 ratio. This was done after aggregating the original split as the training and testing splits provided by Bagnall et al. (2018) had inconsistent ratios across different datasets. After that, random missing observations for each variable were generated, and the modified variables were combined. In each set of cross validation, we used different random seed to investigate the robustness of the methods.

## D    DETAILS OF EXPERIMENTAL SETTINGS

All experiments were performed using a server on Ubuntu 22.04 LTS, equipped with an Intel(R) Xeon(R) Gold 6242 CPU and six NVIDIA A100 40GB GPUs. The source code for our experiments can be accessed at `https://github.com/yongkyung-oh/Stable-Neural-SDEs`.

### D.1    BENCHMARK METHODS

- RNN-based methods: Conventional recurrent neural networks including RNN (Rumelhart et al., 1986; Medsker & Jain, 1999), LSTM (Hochreiter & Schmidhuber, 1997) and GRU (Chung et al., 2014). When dealing with irregularly-sampled or missing data, we apply mean imputation for the missing values. Furthermore, Choi et al. (2016) proposed a model that incorporates lapses in time between observations, alongside the observations themselves. In Che et al. (2018), GRU-D uses a sequence of observed values, missing data indicators, and elapsed times between observations as inputs and learns exponential decay between observations.

- Attention-based methods: Shukla & Marlin (2021) proposed a technique known as Multi-Time Attention Networks (MTAN), which involves learning an embedding of continuous time values and utilizing an attention mechanism. Similarly, Lee et al. (2022) introduced

a method named the Multi-Integration Attention Module (MIAM) for extracting intricate information from irregular time series data with additional attention mechanism.

- Neural ODEs: Neural ODEs (Chen et al., 2018), which are widely utilized for learning continuous latent representations, come in various forms including GRU-ODE (De Brouwer et al., 2019), ODE-RNN (Rubanova et al., 2019), ODE-LSTM (Lechner & Hasani, 2020), Latent-ODE (Rubanova et al., 2019), Augmented-ODE (Dupont et al., 2019), and Attentive co-evolving neural ordinary differential equations (ACE-NODE) (Jhin et al., 2021).

- Neural CDEs: Kidger et al. (2020) introduced the Neural CDE to consider a continuous change of the entire input over time. Neural Rough Differential Equation (Neural RDE) (Morrill et al., 2021), make use of log-signature transformations to directly incorporate time series into the path space. We used depth 2 with mean imputation for the Neural RDE. Attentive Neural Controlled Differential Equation (ANCDE) (Jhin et al., 2023b), employ dual Neural CDEs to compute attention scores. EXtrapolation and InTerpolation-based model (EXIT) (Jhin et al., 2022) and LEArnable Path-based model (LEAP) (Jhin et al., 2023a) utilize an explicit encoder-decoder structure to create the latent control path.

- Neural SDEs: Neural SDEs have been proposed to model genuine random phenomena. SDEs, which represent a logical progression from ODEs, can be applied to the analysis of continuously evolving systems and accommodate uncertainty using drift and diffusion terms (Tzen & Raginsky, 2019; Jia & Benson, 2019; Liu et al., 2019). On the other hand, Li et al. (2020) introduced a method called Latent SDE that uses an adjoint method, which can be considered as an instantaneous analog of the chain rule for solving Neural SDEs. We incorporate Kullback–Leibler (KL) divergence loss for training Latent SDE.

## D.2 NETWORK ARCHITECTURE

There are multiple neural networks in the proposed method, as shown in Equations (12), (13), and (14): $\overline{\gamma}_1(\boldsymbol{z}(t); \theta_\gamma, \theta_\zeta)$, $\overline{\gamma}_2(t, \boldsymbol{z}(t); \theta_\gamma, \theta_\zeta)$, and $\sigma(t; \theta_\sigma)$. The drift terms with control include the mapping function $\zeta : \mathbb{R}_+ \times \mathbb{R}^{d_z} \times \mathbb{R}^{d_x} \to \mathbb{R}^{d_z}$, which uses the concatenated value of the latent value $\boldsymbol{z}(t)$ and the controlled path $\boldsymbol{X}(t)$ as input. Both the drift functions $\overline{\gamma}_1$ and $\overline{\gamma}_2$ are implemented as Multi-layer Perceptrons (MLPs) with `ReLU` activation. The diffusion function $\sigma$ has two options: linear affine and a nonlinear neural network. Our approach involves constraining the SDEs to exhibit diagonal noise, a decision aimed at satisfying the commutativity property as delineated by Rößler (2004). Consequently, our designated networks under Itô -Taylor schemes ensure the pathwise convergence from any given fixed starting point (Kloeden & Neuenkirch, 2007; Li et al., 2020).

In practice, we implemented several design choices to optimize the model's performance and stability. Firstly, we utilized the idea of sinusoidal positional encoding, as proposed by Vaswani et al. (2017), for the time variable, ensuring that each time step possesses a unique encoding. Secondly, we opted for the `tanh` nonlinearity as the final operation for drift, diffusion, and all other vector fields, following the recommendation by Kidger et al. (2020). This choice is intended to prevent the model from experiencing issues related to excessively large values or gradients. Lastly, we employed layer-wise learning rates for the model's final layer (e.g. $\times 100$), facilitating more precise and adaptive learning for the classification task. This approach can result in improved classification outcomes, as each layer can adjust its learning rate according to the complexity of the features it aims to capture.

## D.3 TRAINING STRATEGY

In addressing datasets characterized by irregular sampling or missing values, the initial value of the process is established based on available observations. The initial value, denoted as $x_0$, is interpolated from the observed data $\mathbf{x}$ and subsequently used to define the mapping for $\mathbf{z}(0)$. This methodology is consistent with the approaches outlined by Kidger et al. (2020) and subsequent research.

Subsequently, the latent trajectory value at $\mathbf{z}(T)$ is specifically utilized for designated tasks. This approach ensures that both the estimation of the SDE and the model's output are co-optimized. In our task of classifying time series data, we utilize the cross-entropy loss function, taking the final value of the latent representation $\mathbf{z}(T)$ as input. We employ an MLP, comprising two fully connected layers with `ReLU` activation, to process the extracted feature $\boldsymbol{z} : [0, T]$, as follows:

$$\hat{y} = MLP(\boldsymbol{z}(T); \theta_{MLP}), \tag{30}$$

where $\hat{y}$ is the predicted label of the given time series sample. To mitigate the risk of overfitting and regularize the model, we incorporate a dropout rate of 10%. Also, we employed an early-stopping mechanism, ceasing the training when the validation loss didn't improve for 10 successive epochs. The training approach for our proposed Neural SDEs for classification is outlined in Algorithm 1. The adjoint sensitivity method's gradient computation facilitates the integration of mini-batch algorithms in solving Neural SDEs. For this purpose, python library `torchsde`[8] (Li et al., 2020) is employed, which is adept at handling both naïve Neural SDEs and the proposed Neural SDEs in our research.

---

**Algorithm 1** Train procedure for classification task

---

1: Divide training data into a train set $D_{train}$ and a validation set $D_{val}$. Set the maximum iteration $epoch_{max}$.
2: Initialize the parameters for the control neural network $\theta_\zeta$, the drift term $\theta_\gamma$, and the diffusion term $\theta_\sigma$.
3: Initialize the parameters for the MLP classifier $\theta_{MLP}$.
4: **for** $i = 1$ to $epoch_{max}$ **do**
5:     Train parameters $\boldsymbol{\theta} = [\theta_\zeta, \theta_\gamma, \theta_\sigma, \theta_{MLP}]$ using $D_{train}$ and classification loss $\mathcal{L}(y, \hat{y})$.

$$\arg\min_{\boldsymbol{\theta}} \mathcal{L}(y, \hat{y}).$$

6:     Validate using $D_{val}$ and update parameters.
7:     Find the best parameters $\boldsymbol{\theta}^*$ to minimize validation loss.
8: **end for**
9: **Return** $\theta_\zeta^*, \theta_\gamma^*, \theta_\sigma^*, \theta_{MLP}^*$.

---

## D.4 EMPIRICAL STUDY

Table 7: Analysis of sensitivity for Neural SDEs' depth $T$ and Classification performance utilizing the 'BasicMotions' dataset under each scenario (* denoting the final value)

(a) Neural SDE

| Depth | Regular datasets | Missing datasets (30%) | Missing datasets (50%) | Missing datasets (70%) |
|---|---|---|---|---|
| 10 | 0.417 ± 0.132 | 0.433 ± 0.070 | 0.467 ± 0.095 | 0.417 ± 0.132 |
| 50 | 0.383 ± 0.112 | 0.417 ± 0.144 | 0.417 ± 0.118 | 0.383 ± 0.162 |
| 90 | 0.483 ± 0.124 | 0.383 ± 0.095 | 0.483 ± 0.091 | 0.383 ± 0.151 |
| 100* | 0.417 ± 0.000 | 0.333 ± 0.059 | 0.500 ± 0.059 | 0.467 ± 0.095 |

(b) Neural LSDE

| Depth | Regular datasets | Missing datasets (30%) | Missing datasets (50%) | Missing datasets (70%) |
|---|---|---|---|---|
| 10 | 0.433 ± 0.137 | 0.417 ± 0.118 | 0.300 ± 0.095 | 0.450 ± 0.173 |
| 50 | 1.000 ± 0.000 | 0.967 ± 0.046 | 0.950 ± 0.075 | 0.867 ± 0.046 |
| 90 | 1.000 ± 0.000 | 0.967 ± 0.046 | 0.967 ± 0.046 | 0.917 ± 0.000 |
| 100* | 1.000 ± 0.000 | 1.000 ± 0.000 | 1.000 ± 0.000 | 0.983 ± 0.037 |

(c) Neural LNSDE

| Depth | Regular datasets | Missing datasets (30%) | Missing datasets (50%) | Missing datasets (70%) |
|---|---|---|---|---|
| 10 | 0.333 ± 0.102 | 0.400 ± 0.109 | 0.450 ± 0.139 | 0.367 ± 0.139 |
| 50 | 1.000 ± 0.000 | 0.967 ± 0.046 | 0.933 ± 0.070 | 0.900 ± 0.070 |
| 90 | 1.000 ± 0.000 | 0.983 ± 0.037 | 0.983 ± 0.037 | 0.933 ± 0.070 |
| 100* | 1.000 ± 0.000 | 1.000 ± 0.000 | 1.000 ± 0.000 | 0.983 ± 0.037 |

(d) Neural GSDE

| Depth | Regular datasets | Missing datasets (30%) | Missing datasets (50%) | Missing datasets (70%) |
|---|---|---|---|---|
| 10 | 0.483 ± 0.124 | 0.417 ± 0.102 | 0.400 ± 0.070 | 0.417 ± 0.144 |
| 50 | 0.983 ± 0.037 | 0.950 ± 0.046 | 0.933 ± 0.070 | 0.883 ± 0.095 |
| 90 | 1.000 ± 0.000 | 0.967 ± 0.046 | 0.983 ± 0.037 | 0.917 ± 0.059 |
| 100* | 1.000 ± 0.000 | 0.983 ± 0.037 | 0.983 ± 0.037 | 0.983 ± 0.037 |

---

[8] https://github.com/google-research/torchsde

**Impact of the Neural SDEs' depth $T$.** Theorems 3.5 and 3.6 establish non-asymptotic upper bounds on the variance between output distributions of original and perturbed input data. To empirically verify this hypothesis, the impact of varying $T$ on Neural SDEs' performance is examined.

Table 7 illustrates the classification results obtained from the 'BasicMotions' dataset at various depth levels. In alignment with our theoretical expectations, it is observed that the model's efficacy consolidates and its robustness intensifies as the depth parameter $T$ is increased, substantiating the durability of the proposed methods in scenarios of varying depths.

Table 8: Classification performance on the 'BasicMotions' dataset at a 50% missing rate using different solvers: the explicit Euler, the Milstein, and the Stochastic Runge-Kutta (SRK) (Average taken over five iterations. Runtime, in seconds, for 100 epochs without early-stopping.)

| Methods | Solvers | Accuracy | Runtime |
|---|---|---|---|
| **Neural SDE** | **Euler** | 0.500 ± 0.059 | 87.4 ± 0.3 |
| | Milstein | 0.483 ± 0.091 | 156.9 ± 0.9 |
| | SRK | 0.500 ± 0.059 | 749.4 ± 3.4 |
| **Neural LSDE** | **Euler** | 1.000 ± 0.000 | 89.4 ± 0.5 |
| | Milstein | 0.983 ± 0.037 | 112.6 ± 1.0 |
| | SRK | 1.000 ± 0.000 | 748.5 ± 2.4 |
| **Neural LNSDE** | **Euler** | 1.000 ± 0.000 | 95.0 ± 0.4 |
| | Milstein | 0.983 ± 0.037 | 135.3 ± 0.8 |
| | SRK | 0.983 ± 0.037 | 822.5 ± 2.2 |
| **Neural GSDE** | **Euler** | 0.983 ± 0.037 | 96.4 ± 0.6 |
| | Milstein | 0.983 ± 0.037 | 135.7 ± 1.1 |
| | SRK | 0.967 ± 0.046 | 839.1 ± 4.1 |

Table 9: Classification performance on the 'BasicMotions' dataset at a 50% missing rate (Average taken over five iterations. Runtime, in seconds, for 100 epochs without early-stopping.)

| Methods | Accuracy | Runtime |
|---|---|---|
| RNN | 0.600 ± 0.070 | 2.3 ± 0.1 |
| LSTM | 0.700 ± 0.095 | 2.8 ± 0.9 |
| GRU | 0.900 ± 0.091 | 3.0 ± 0.0 |
| GRU-$\Delta t$ | 0.967 ± 0.046 | 34.4 ± 0.3 |
| GRU-D | 0.933 ± 0.070 | 42.4 ± 0.3 |
| MTAN | 0.717 ± 0.247 | 6.2 ± 0.4 |
| MIAM | 0.933 ± 0.109 | 41.7 ± 0.6 |
| GRU-ODE | 0.983 ± 0.037 | 519.8 ± 0.3 |
| ODE-RNN | 1.000 ± 0.000 | 113.3 ± 0.2 |
| ODE-LSTM | 0.717 ± 0.126 | 73.0 ± 0.7 |
| Neural CDE | 0.983 ± 0.037 | 102.5 ± 0.3 |
| Neural RDE | 0.717 ± 0.046 | 93.8 ± 0.4 |
| ANCDE | 0.983 ± 0.037 | 360.5 ± 0.5 |
| EXIT | 0.417 ± 0.102 | 334.5 ± 0.3 |
| LEAP | 0.317 ± 0.124 | 163.0 ± 0.6 |
| Latent SDE | 0.350 ± 0.109 | 238.5 ± 0.6 |
| **Neural SDE** | 0.500 ± 0.059 | 87.4 ± 0.3 |
| **Neural LSDE** | 1.000 ± 0.000 | 89.4 ± 0.5 |
| **Neural LNSDE** | 1.000 ± 0.000 | 95.0 ± 0.4 |
| **Neural GSDE** | 0.983 ± 0.037 | 96.4 ± 0.6 |

**Choice of solver.** We considered three distinct numerical solvers for Neural SDEs: the explicit Euler-Maruyama method, the Milstein method, and the Stochastic Runge-Kutta (SRK) method. Each solver presents varying degrees of convergence and computational efficiency: the Euler method demonstrates a convergence order of 0.5, while the Milstein and SRK methods exhibit higher orders of 1.0 and 1.5, respectively, indicating progressively enhanced accuracy.

For our study involving high-dimensional SDEs, the explicit Euler method was selected due to its superior computational efficiency, despite its lower accuracy, in comparison to the more computationally intensive Milstein and SRK methods. This choice was motivated by the necessity to manage the computational demands of high-dimensional data effectively. As shown in Table 8, this selection is further validated by the average runtimes observed using the 'BasicMotions' dataset.

Recognizing the potential concerns about numerical stability in explicit numerical solutions, our Neural SDE models were carefully crafted to ensure robust performance. This design forethought is crucial in maintaining numerical stability throughout the computations utilizing the explicit Euler method. Table 9 provides an investigation into both the classification accuracy and the average runtime, reflecting the efficiency and effectiveness of our chosen methodology versus benchmarks.

# E    DETAILED RESULTS OF THE BENCHMARK DATASETS

For the interpolation task, we employed the dataset pipeline recommended for **PhysioNet Mortality** (Silva et al., 2012), as outlined by Shukla & Marlin (2021), with additional resources available in its GitHub repository[9]. The forecasting task utilized the **MuJoCo dataset** (Tassa et al., 2018), following the experimental procedures and resources found in Jhin et al. (2023b) and its corresponding GitHub repository[10]. For the datasets **PhysioNet Sepsis** (Reyna et al., 2019), and **Speech**

---

[9]https://github.com/reml-lab/mTAN
[10]https://github.com/sheoyon-jhin/ANCDE

**Commands** (Warden, 2018), we adopted the preprocessing and experimental protocols detailed in Kidger et al. (2020) and its corresponding Github repository[11]. We recommend referring to the original paper for further details regarding the data and experiment protocols.

**PhysioNet Mortality.** We compare the proposed techniques with the benchmark method delineated in Shukla & Marlin (2021). The following models are considered: RNN-VAE model (Chen et al., 2018), employs a Variational AutoEncoder (VAE) framework wherein both the encoder and decoder are instantiated as conventional RNN architectures. L-ODE-RNN model (Chen et al., 2018), is a Latent ODE variant that utilizes an RNN encoder and a Neural ODE decoder. L-ODE-ODE (Rubanova et al., 2019), is another Latent ODE variant, but both its encoder and decoder are based on ODE-RNN and Neural ODE respectively. MTAN (Shukla & Marlin, 2021), integrates a time attention mechanism alongside Bidirectional RNNs to encapsulate temporal features and interpolate data.

Table 10: Results of hyperparameter tuning for 'PhysioNet Mortality': Mean Squared Error (MSE, scaled by $10^{-3}$) for interpolation tasks across various observation levels

(a) Neural SDE

| $n_l$ | $n_h$ | Observed % | | | | |
|---|---|---|---|---|---|---|
| | | 50% | 60% | 70% | 80% | 90% |
| 1 | 16 | 8.900 | 8.892 | 8.666 | 8.593 | 8.359 |
| | 32 | 8.953 | 8.785 | 8.719 | 8.603 | 8.377 |
| | 64 | 8.739 | 8.718 | 8.613 | 8.392 | 8.374 |
| | 128 | 8.770 | 8.686 | 8.560 | 8.464 | 8.296 |
| 2 | 16 | 8.901 | 8.835 | 8.706 | 8.624 | 8.445 |
| | 32 | 8.861 | 8.858 | 8.669 | 8.550 | 8.446 |
| | 64 | 8.842 | 8.829 | 8.668 | 8.518 | 8.362 |
| | 128 | 8.826 | 8.705 | 8.526 | 8.430 | 8.305 |
| 3 | 16 | 8.899 | 8.782 | 8.652 | 8.517 | 8.427 |
| | 32 | 8.812 | 8.829 | 8.607 | 8.470 | 8.314 |
| | 64 | 8.687 | **8.481** | **8.471** | 8.325 | **8.147** |
| | 128 | 8.696 | 8.652 | 8.533 | 8.412 | 8.312 |
| 4 | 16 | 8.902 | 8.690 | 8.715 | 8.521 | 8.473 |
| | 32 | 8.854 | 8.657 | 8.571 | 8.398 | 8.367 |
| | 64 | 8.788 | 8.658 | 8.592 | 8.459 | 8.263 |
| | 128 | **8.592** | 8.591 | 8.540 | **8.318** | 8.252 |

(b) Neural LSDE

| $n_l$ | $n_h$ | Observed % | | | | |
|---|---|---|---|---|---|---|
| | | 50% | 60% | 70% | 80% | 90% |
| 1 | 16 | 5.315 | 5.017 | 4.765 | 4.654 | 4.446 |
| | 32 | 5.081 | 5.039 | 4.596 | 4.632 | 3.732 |
| | 64 | 4.623 | 4.521 | 4.437 | 4.364 | 3.347 |
| | 128 | 3.822 | 3.658 | 3.516 | 3.355 | 3.098 |
| 2 | 16 | 5.399 | 5.286 | 5.153 | 5.104 | 4.336 |
| | 32 | 4.711 | 4.520 | 4.332 | 4.097 | 4.000 |
| | 64 | 4.200 | 4.066 | 3.827 | 3.632 | 3.351 |
| | 128 | 4.310 | 4.167 | 4.130 | 4.057 | 3.106 |
| 3 | 16 | 5.585 | 5.367 | 5.295 | 5.242 | 4.683 |
| | 32 | 4.806 | 4.567 | 4.220 | 3.988 | 3.789 |
| | 64 | 4.123 | 4.015 | 3.631 | 3.441 | 3.340 |
| | 128 | **3.793** | 3.645 | **3.350** | **3.209** | **3.089** |
| 4 | 16 | 5.223 | 5.296 | 4.964 | 4.811 | 4.665 |
| | 32 | 4.682 | 4.751 | 4.107 | 4.046 | 3.848 |
| | 64 | 4.139 | 3.861 | 3.756 | 3.456 | 3.380 |
| | 128 | 3.799 | **3.584** | 3.457 | 3.262 | 3.111 |

(c) Neural LNSDE

| $n_l$ | $n_h$ | Observed % | | | | |
|---|---|---|---|---|---|---|
| | | 50% | 60% | 70% | 80% | 90% |
| 1 | 16 | 5.367 | 5.362 | 4.914 | 4.516 | 4.224 |
| | 32 | 4.980 | 4.309 | 4.674 | 4.616 | 3.766 |
| | 64 | 4.490 | 4.591 | 4.356 | 4.531 | 3.503 |
| | 128 | **3.800** | 3.763 | 3.492 | 3.293 | 3.060 |
| 2 | 16 | 5.297 | 5.374 | 4.842 | 5.030 | 4.703 |
| | 32 | 4.645 | 4.628 | 4.531 | 4.197 | 4.060 |
| | 64 | 4.171 | 4.043 | 3.838 | 3.705 | 3.303 |
| | 128 | 4.309 | 4.199 | 4.130 | 3.535 | 3.162 |
| 3 | 16 | 5.455 | 5.411 | 5.074 | 5.152 | 4.766 |
| | 32 | 4.726 | 4.383 | 4.052 | 3.963 | 4.040 |
| | 64 | 4.082 | 3.905 | 3.656 | 3.457 | 3.395 |
| | 128 | 3.829 | **3.600** | **3.353** | **3.193** | **3.040** |
| 4 | 16 | 5.365 | 5.119 | 4.809 | 4.760 | 4.608 |
| | 32 | 4.674 | 4.180 | 4.260 | 3.945 | 4.064 |
| | 64 | 4.114 | 3.823 | 3.689 | 3.458 | 3.398 |
| | 128 | 3.808 | 3.617 | 3.405 | 3.269 | 3.154 |

(d) Neural GSDE

| $n_l$ | $n_h$ | Observed % | | | | |
|---|---|---|---|---|---|---|
| | | 50% | 60% | 70% | 80% | 90% |
| 1 | 16 | 5.408 | 5.245 | 5.159 | 5.090 | 4.499 |
| | 32 | 5.093 | 4.961 | 4.320 | 4.888 | 3.968 |
| | 64 | 4.699 | 4.191 | 4.470 | 4.464 | 3.477 |
| | 128 | 4.149 | 3.882 | 4.219 | 4.072 | 3.195 |
| 2 | 16 | 5.508 | 5.491 | 5.090 | 4.650 | 4.541 |
| | 32 | 4.832 | 4.766 | 4.595 | 4.196 | 4.094 |
| | 64 | 4.215 | 4.078 | 3.936 | 3.702 | 3.427 |
| | 128 | 4.369 | 4.274 | 4.222 | 3.952 | 3.158 |
| 3 | 16 | 5.591 | 5.498 | 5.311 | 4.896 | 4.754 |
| | 32 | 4.684 | 4.345 | 4.224 | 4.098 | 3.869 |
| | 64 | 4.273 | 4.046 | 3.760 | 3.663 | 3.486 |
| | 128 | 3.863 | **3.661** | 3.453 | 3.296 | **3.089** |
| 4 | 16 | 5.449 | 4.918 | 4.911 | 4.780 | 4.649 |
| | 32 | 4.672 | 4.496 | 4.377 | 4.145 | 3.902 |
| | 64 | 4.242 | 3.966 | 3.753 | 3.555 | 3.497 |
| | 128 | **3.824** | 3.667 | 3.493 | **3.287** | 3.118 |

For the proposed methodology, the training process spans 300 epochs, employing a batch size of 64 and a learning rate of 0.001. To train our models on a dataset comprising irregularly sampled time series, we adopt a strategy from Shukla & Marlin (2021). This involves the modified VAE training method, where we optimize a normalized variational lower bound of the log marginal likelihood,

---
[11] https://github.com/patrick-kidger/NeuralCDE

grounded on the evidence lower bound (ELBO). Hyperparameter optimization is conducted through a grid search, focusing on the number of layers in vector field $n_l$ of $\{16, 32, 64, 128\}$ and hidden vector dimensions $n_h$ of $\{16, 32, 64, 128\}$. The optimal hyperparameters for the proposed methods are remarked in bold within Table 10, encompassing observed data ranging from 50% to 90%.

**MuJoCo.** We evaluate the proposed model against the performance provided on Jhin et al. (2023b). The methods we compared include GRU-$\Delta t$ (Choi et al., 2016), GRU-D (Che et al., 2018), GRU-ODE (De Brouwer et al., 2019), ODE-RNN (Rubanova et al., 2019), Latent-ODE (Rubanova et al., 2019), Augmented-ODE (Dupont et al., 2019), ACE-NODE (Jhin et al., 2021), Neural CDE (Kidger et al., 2020), ANCDE (Jhin et al., 2023b), EXIT (Jhin et al., 2022), and LEAP (Jhin et al., 2023a).

Table 11: Results of hyperparameter tuning for 'MuJoCo' with regular dataset

| $n_l$ | $n_h$ | Neural SDE | | Neural LSDE | | Neural LNSDE | | Neural GSDE | |
| | | Test MSE | Memory | Test MSE | Memory | Test MSE | Memory | Test MSE | Memory |
|---|---|---|---|---|---|---|---|---|---|
| 1 | 16 | 0.057 ± 0.009 | 66 | 0.031 ± 0.001 | 72 | 0.032 ± 0.001 | 78 | 0.031 ± 0.004 | 87 |
| | 32 | 0.040 ± 0.001 | 118 | 0.022 ± 0.001 | 127 | 0.021 ± 0.002 | 139 | 0.019 ± 0.001 | 157 |
| | 64 | 0.041 ± 0.008 | 218 | 0.018 ± 0.001 | 237 | 0.016 ± 0.001 | 261 | 0.016 ± 0.001 | 298 |
| | 128 | 0.046 ± 0.002 | 432 | 0.017 ± 0.001 | 460 | 0.016 ± 0.001 | 508 | **0.013 ± 0.001** | 582 |
| 2 | 16 | 0.054 ± 0.011 | 69 | 0.041 ± 0.001 | 75 | 0.033 ± 0.002 | 81 | 0.034 ± 0.000 | 90 |
| | 32 | 0.038 ± 0.003 | 124 | 0.024 ± 0.002 | 133 | 0.024 ± 0.002 | 141 | 0.025 ± 0.001 | 160 |
| | 64 | 0.032 ± 0.000 | 234 | 0.017 ± 0.001 | 249 | 0.017 ± 0.001 | 273 | 0.018 ± 0.001 | 306 |
| | 128 | 0.031 ± 0.002 | 449 | **0.013 ± 0.000** | 485 | **0.012 ± 0.000** | 533 | 0.015 ± 0.001 | 607 |
| 3 | 16 | 0.054 ± 0.012 | 72 | 0.043 ± 0.000 | 78 | 0.036 ± 0.003 | 84 | 0.039 ± 0.003 | 89 |
| | 32 | 0.061 ± 0.012 | 130 | 0.030 ± 0.000 | 139 | 0.029 ± 0.000 | 151 | 0.029 ± 0.001 | 162 |
| | 64 | 0.031 ± 0.001 | 246 | 0.023 ± 0.000 | 261 | 0.023 ± 0.002 | 286 | 0.022 ± 0.003 | 310 |
| | 128 | **0.028 ± 0.001** | 474 | 0.018 ± 0.001 | 509 | 0.017 ± 0.001 | 558 | 0.018 ± 0.001 | 615 |
| 4 | 16 | 0.065 ± 0.004 | 75 | 0.046 ± 0.000 | 81 | 0.044 ± 0.005 | 86 | 0.036 ± 0.001 | 92 |
| | 32 | 0.061 ± 0.010 | 136 | 0.035 ± 0.002 | 145 | 0.037 ± 0.001 | 157 | 0.031 ± 0.000 | 169 |
| | 64 | 0.031 ± 0.003 | 259 | 0.027 ± 0.000 | 274 | 0.028 ± 0.000 | 298 | 0.027 ± 0.000 | 322 |
| | 128 | 0.031 ± 0.004 | 507 | 0.023 ± 0.002 | 534 | 0.023 ± 0.001 | 583 | 0.023 ± 0.001 | 632 |

For the proposed methodology, we spanned 500 epochs with a batch size of 1024 and a learning rate set at 0.001. We performed hyperparameter optimization using a grid search, focusing specifically on the number of layers in vector field $n_l$ of $\{16, 32, 64, 128\}$ and hidden vector dimensions $n_h$ of $\{16, 32, 64, 128\}$. The optimal hyperparameters, as identified for our methods, are highlighted in bold in Table 11 for regular datasets. For scenarios with 30%, 50%, and 70% missing data, we applied the same hyperparameters.

Table 12: Forecasting performance versus percent observed time points on MuJoCo

| Methods | Test MSE | | | | Memory Usage (MB) |
| | Regular | 30% dropped | 50% dropped | 70% dropped | |
|---|---|---|---|---|---|
| GRU-$\Delta t$ | 0.223 ± 0.020 | 0.198 ± 0.036 | 0.193 ± 0.015 | 0.196 ± 0.028 | 533 |
| GRU-D | 0.578 ± 0.042 | 0.608 ± 0.032 | 0.587 ± 0.039 | 0.579 ± 0.052 | 569 |
| GRU-ODE | 0.856 ± 0.016 | 0.857 ± 0.015 | 0.852 ± 0.015 | 0.861 ± 0.015 | 146 |
| ODE-RNN | 0.328 ± 0.225 | 0.274 ± 0.213 | 0.237 ± 0.110 | 0.267 ± 0.217 | 115 |
| Latent-ODE | 0.029 ± 0.011 | 0.056 ± 0.001 | 0.055 ± 0.004 | 0.058 ± 0.003 | 314 |
| Augmented-ODE | 0.055 ± 0.004 | 0.056 ± 0.004 | 0.057 ± 0.005 | 0.057 ± 0.005 | 286 |
| ACE-NODE | 0.039 ± 0.003 | 0.053 ± 0.007 | 0.053 ± 0.005 | 0.052 ± 0.006 | 423 |
| NCDE | 0.028 ± 0.002 | 0.027 ± 0.000 | 0.027 ± 0.001 | 0.026 ± 0.001 | 52 |
| ANCDE | 0.026 ± 0.001 | 0.025 ± 0.001 | 0.025 ± 0.001 | 0.024 ± 0.001 | 79 |
| EXIT | 0.026 ± 0.000 | 0.025 ± 0.004 | 0.026 ± 0.000 | 0.026 ± 0.001 | 127 |
| LEAP | 0.022 ± 0.002 | 0.022 ± 0.001 | 0.022 ± 0.002 | 0.022 ± 0.001 | 144 |
| **Neural SDE** | 0.028 ± 0.004 | 0.029 ± 0.001 | 0.029 ± 0.001 | 0.027 ± 0.000 | 234 |
| **Neural LSDE** | 0.013 ± 0.000 | 0.014 ± 0.001 | 0.014 ± 0.000 | **0.013 ± 0.001** | 249 |
| **Neural LNSDE** | **0.012 ± 0.001** | 0.014 ± 0.001 | 0.014 ± 0.001 | 0.014 ± 0.000 | 273 |
| **Neural GSDE** | 0.013 ± 0.001 | **0.013 ± 0.001** | **0.013 ± 0.000** | 0.014 ± 0.000 | 306 |

Table 12 presents the forecasting performance across varying data drop ratios. Our methods consistently demonstrate lower MSE scores, indicating their superior forecasting capabilities. However, the proposed methods demand higher computational resources due to the complex architecture.

**PhysioNet Sepsis.** We evaluate the proposed model against the performance provided on Jhin et al. (2023b). The methods we compared include GRU-$\Delta t$ (Choi et al., 2016), GRU-D (Che et al., 2018), GRU-ODE (De Brouwer et al., 2019), ODE-RNN (Rubanova et al., 2019), Latent-ODE (Rubanova et al., 2019), ACE-NODE (Jhin et al., 2021), Neural CDE (Kidger et al., 2020), and ANCDE (Jhin et al., 2023b). We examine the results both with and without the observational intensity (OI), which is determined by attaching a mask indicating whether an observation was made or not to each input.

Table 13: Results of hyperparameter tuning for 'PhysioNet Sepsis' with observation intensity

| $n_l$ | $n_h$ | Neural SDE | | Neural LSDE | | Neural LNSDE | | Neural GSDE | |
| | | Test AUROC | Memory | Test AUROC | Memory | Test AUROC | Memory | Test AUROC | Memory |
|---|---|---|---|---|---|---|---|---|---|
| 1 | 16 | $0.792 \pm 0.008$ | 304 | $0.907 \pm 0.003$ | 314 | $0.910 \pm 0.002$ | 340 | $0.909 \pm 0.002$ | 338 |
| | 32 | $0.780 \pm 0.006$ | 369 | $0.904 \pm 0.002$ | 353 | $0.899 \pm 0.003$ | 366 | $0.902 \pm 0.004$ | 369 |
| | 64 | $0.790 \pm 0.006$ | 442 | $0.895 \pm 0.002$ | 421 | $0.881 \pm 0.007$ | 485 | $0.894 \pm 0.007$ | 509 |
| | 128 | $0.769 \pm 0.008$ | 736 | $0.890 \pm 0.010$ | 747 | $0.870 \pm 0.015$ | 758 | $0.883 \pm 0.007$ | 924 |
| 2 | 16 | $0.793 \pm 0.004$ | 338 | $0.903 \pm 0.005$ | 344 | $\mathbf{0.911 \pm 0.002}$ | 341 | $0.907 \pm 0.001$ | 298 |
| | 32 | $0.786 \pm 0.010$ | 379 | $\mathbf{0.909 \pm 0.004}$ | 373 | $0.903 \pm 0.005$ | 351 | $0.903 \pm 0.002$ | 332 |
| | 64 | $0.763 \pm 0.005$ | 486 | $0.907 \pm 0.003$ | 489 | $0.898 \pm 0.006$ | 522 | $0.900 \pm 0.007$ | 536 |
| | 128 | $0.769 \pm 0.010$ | 802 | $0.906 \pm 0.004$ | 818 | $0.867 \pm 0.007$ | 860 | $0.882 \pm 0.009$ | 965 |
| 3 | 16 | $0.793 \pm 0.003$ | 346 | $0.905 \pm 0.001$ | 341 | $0.909 \pm 0.002$ | 338 | $0.904 \pm 0.006$ | 281 |
| | 32 | $\mathbf{0.799 \pm 0.007}$ | 368 | $0.908 \pm 0.004$ | 359 | $0.908 \pm 0.003$ | 341 | $0.906 \pm 0.002$ | 347 |
| | 64 | $0.776 \pm 0.006$ | 491 | $0.901 \pm 0.006$ | 481 | $0.902 \pm 0.007$ | 495 | $0.901 \pm 0.002$ | 511 |
| | 128 | $0.776 \pm 0.006$ | 774 | $0.902 \pm 0.005$ | 822 | $0.895 \pm 0.004$ | 863 | $0.895 \pm 0.003$ | 936 |
| 4 | 16 | $0.782 \pm 0.011$ | 333 | $0.902 \pm 0.007$ | 299 | $0.906 \pm 0.005$ | 341 | $0.893 \pm 0.007$ | 297 |
| | 32 | $0.784 \pm 0.012$ | 376 | $0.902 \pm 0.001$ | 355 | $0.908 \pm 0.003$ | 345 | $0.906 \pm 0.004$ | 362 |
| | 64 | $0.780 \pm 0.005$ | 488 | $0.907 \pm 0.004$ | 476 | $0.899 \pm 0.005$ | 541 | $\mathbf{0.909 \pm 0.001}$ | 588 |
| | 128 | $0.764 \pm 0.013$ | 844 | $0.900 \pm 0.002$ | 785 | $0.900 \pm 0.004$ | 896 | $0.905 \pm 0.001$ | 962 |

Table 14: Results of hyperparameter tuning for 'PhysioNet Sepsis' without observation intensity

| $n_l$ | $n_h$ | Neural SDE | | Neural LSDE | | Neural LNSDE | | Neural GSDE | |
| | | Test AUROC | Memory | Test AUROC | Memory | Test AUROC | Memory | Test AUROC | Memory |
|---|---|---|---|---|---|---|---|---|---|
| 1 | 16 | $0.782 \pm 0.008$ | 171 | $0.867 \pm 0.004$ | 183 | $0.873 \pm 0.006$ | 173 | $0.875 \pm 0.002$ | 181 |
| | 32 | $0.787 \pm 0.008$ | 216 | $0.869 \pm 0.006$ | 256 | $0.868 \pm 0.010$ | 248 | $0.873 \pm 0.003$ | 267 |
| | 64 | $0.790 \pm 0.003$ | 353 | $0.870 \pm 0.004$ | 353 | $0.832 \pm 0.012$ | 388 | $0.852 \pm 0.011$ | 402 |
| | 128 | $0.773 \pm 0.005$ | 659 | $0.829 \pm 0.010$ | 712 | $0.775 \pm 0.020$ | 783 | $0.809 \pm 0.009$ | 868 |
| 2 | 16 | $0.795 \pm 0.009$ | 191 | $0.872 \pm 0.007$ | 181 | $0.874 \pm 0.004$ | 188 | $0.880 \pm 0.002$ | 184 |
| | 32 | $0.776 \pm 0.007$ | 233 | $0.867 \pm 0.002$ | 245 | $0.877 \pm 0.006$ | 245 | $0.880 \pm 0.005$ | 260 |
| | 64 | $0.782 \pm 0.012$ | 394 | $0.866 \pm 0.004$ | 447 | $0.865 \pm 0.007$ | 395 | $0.870 \pm 0.006$ | 467 |
| | 128 | $0.775 \pm 0.006$ | 592 | $0.867 \pm 0.006$ | 761 | $0.815 \pm 0.011$ | 772 | $0.854 \pm 0.007$ | 833 |
| 3 | 16 | $0.787 \pm 0.009$ | 198 | $0.865 \pm 0.001$ | 181 | $0.869 \pm 0.006$ | 191 | $0.872 \pm 0.006$ | 164 |
| | 32 | $\mathbf{0.797 \pm 0.006}$ | 240 | $0.867 \pm 0.006$ | 248 | $0.878 \pm 0.006$ | 261 | $\mathbf{0.884 \pm 0.002}$ | 280 |
| | 64 | $0.783 \pm 0.007$ | 386 | $0.868 \pm 0.004$ | 420 | $0.863 \pm 0.005$ | 443 | $0.878 \pm 0.001$ | 500 |
| | 128 | $0.764 \pm 0.004$ | 704 | $0.837 \pm 0.015$ | 802 | $0.846 \pm 0.011$ | 780 | $0.870 \pm 0.004$ | 968 |
| 4 | 16 | $0.762 \pm 0.005$ | 184 | $0.859 \pm 0.007$ | 189 | $0.865 \pm 0.003$ | 178 | $0.868 \pm 0.007$ | 186 |
| | 32 | $0.781 \pm 0.006$ | 254 | $0.862 \pm 0.009$ | 266 | $0.872 \pm 0.002$ | 284 | $0.877 \pm 0.004$ | 304 |
| | 64 | $0.777 \pm 0.005$ | 396 | $\mathbf{0.879 \pm 0.008}$ | 436 | $\mathbf{0.881 \pm 0.002}$ | 445 | $0.878 \pm 0.003$ | 461 |
| | 128 | $0.752 \pm 0.017$ | 789 | $0.866 \pm 0.006$ | 747 | $0.859 \pm 0.005$ | 850 | $0.875 \pm 0.003$ | 994 |

In the proposed method, we train for 200 epochs with a batch size of 1024 and a learning rate of 0.001. The hyperparameters are optimized by grid search in the number of layers in vector field $n_l$ of $\{16, 32, 64, 128\}$ and hidden vector dimensions $n_h$ of $\{16, 32, 64, 128\}$. The best hyperparameters are highlighted with bold font in Table 13 and 14. Due to varying input channel counts in each scenario, the best hyperparameters might differ. The interplay between hyperparameters and data traits is vital for model efficacy and adaptability.

**Speech Commands.** We evaluate the proposed model against the performance provided on Jhin et al. (2023a). The methods we compared include RNN (Medsker & Jain, 1999), LSTM (Hochreiter & Schmidhuber, 1997), GRU (Chung et al., 2014), GRU-$\Delta t$ (Choi et al., 2016), GRU-D (Che et al., 2018), GRU-ODE (De Brouwer et al., 2019), ODE-RNN (Rubanova et al., 2019), Latent-ODE (Rubanova et al., 2019), Augmented-ODE (Dupont et al., 2019), ACE-NODE (Jhin et al., 2021), Neural CDE (Kidger et al., 2020), ANCDE (Jhin et al., 2023b), and LEAP (Jhin et al., 2023a)

Table 15: Results of hyperparameter tuning for 'Speech Commands'

| $n_l$ | $n_h$ | Neural SDE | | Neural LSDE | | Neural LNSDE | | Neural GSDE | |
|---|---|---|---|---|---|---|---|---|---|
| | | Test Accuracy | Memory | Test Accuracy | Memory | Test Accuracy | Memory | Test Accuracy | Memory |
| 1 | 16 | $0.105 \pm 0.001$ | 238 | $0.732 \pm 0.013$ | 246 | $0.749 \pm 0.009$ | 250 | $0.739 \pm 0.009$ | 261 |
| | 32 | $\mathbf{0.110 \pm 0.005}$ | 309 | $0.837 \pm 0.011$ | 347 | $0.846 \pm 0.004$ | 352 | $0.837 \pm 0.005$ | 436 |
| | 64 | $0.105 \pm 0.001$ | 521 | $0.866 \pm 0.008$ | 543 | $0.892 \pm 0.005$ | 656 | $0.876 \pm 0.005$ | 698 |
| | 128 | $0.104 \pm 0.002$ | 880 | $0.866 \pm 0.037$ | 982 | $\mathbf{0.924 \pm 0.000}$ | 1164 | $0.913 \pm 0.003$ | 1399 |
| 2 | 16 | $0.106 \pm 0.003$ | 213 | $0.839 \pm 0.016$ | 259 | $0.757 \pm 0.007$ | 198 | $0.746 \pm 0.002$ | 252 |
| | 32 | $0.105 \pm 0.002$ | 315 | $0.886 \pm 0.009$ | 353 | $0.859 \pm 0.003$ | 377 | $0.848 \pm 0.002$ | 476 |
| | 64 | $0.109 \pm 0.005$ | 535 | $0.900 \pm 0.006$ | 599 | $0.903 \pm 0.003$ | 649 | $0.895 \pm 0.002$ | 734 |
| | 128 | $0.109 \pm 0.004$ | 995 | $0.910 \pm 0.005$ | 1142 | $0.915 \pm 0.003$ | 1211 | $0.909 \pm 0.003$ | 1431 |
| 3 | 16 | $0.108 \pm 0.004$ | 215 | $0.847 \pm 0.009$ | 237 | $0.700 \pm 0.020$ | 240 | $0.711 \pm 0.005$ | 263 |
| | 32 | $0.107 \pm 0.002$ | 343 | $0.904 \pm 0.001$ | 398 | $0.838 \pm 0.007$ | 456 | $0.844 \pm 0.003$ | 467 |
| | 64 | $0.103 \pm 0.002$ | 577 | $0.919 \pm 0.003$ | 557 | $0.898 \pm 0.006$ | 731 | $0.895 \pm 0.004$ | 843 |
| | 128 | $0.107 \pm 0.002$ | 1097 | $\mathbf{0.927 \pm 0.004}$ | 1187 | $0.922 \pm 0.001$ | 1277 | $\mathbf{0.913 \pm 0.001}$ | 1565 |
| 4 | 16 | $0.108 \pm 0.004$ | 233 | $0.847 \pm 0.013$ | 243 | $0.383 \pm 0.069$ | 267 | $0.632 \pm 0.019$ | 276 |
| | 32 | $0.107 \pm 0.001$ | 363 | $0.900 \pm 0.002$ | 387 | $0.782 \pm 0.031$ | 423 | $0.800 \pm 0.016$ | 482 |
| | 64 | $0.106 \pm 0.001$ | 573 | $0.916 \pm 0.003$ | 630 | $0.887 \pm 0.009$ | 692 | $0.872 \pm 0.007$ | 812 |
| | 128 | $0.106 \pm 0.002$ | 1157 | $0.926 \pm 0.005$ | 1055 | $0.912 \pm 0.004$ | 1372 | $0.899 \pm 0.005$ | 1603 |

In the proposed method, we train for 200 epochs with a batch size of 1024 and a learning rate of 0.001. The hyperparameters are optimized by grid search in the number of layers in vector field $n_l$ of $\{16, 32, 64, 128\}$ and hidden vector dimensions $n_h$ of $\{16, 32, 64, 128\}$. The best hyperparameters are highlighted with bold font in Table 15. Relative to the naïve Neural SDE, our suggested approaches deliver enhanced results through a more intricate model framework.

**Robustness to missing data.** We utilized the Python library `torchcde`[12] (Kidger et al., 2020; Morrill et al., 2021) for the interpolation scheme and the Python library `torchsde`[13] (Li et al., 2020) to solve Neural SDEs for both the Neural SDE and the proposed methods. This study utilized the original Neural CDE source code[14] (Kidger et al., 2020) for benchmark methods including GRU-$\Delta t$, GRU-D, GRU-ODE, ODE-RNN, and Neural CDE. Additionally, we employed ODE-LSTM using its original source code[15] (Lechner & Hasani, 2020), as well as Neural RDE[16] (Morrill et al., 2021), ANCDE[17] (Jhin et al., 2023b), EXIT[18] (Jhin et al., 2022), and LEAP[19] (Jhin et al., 2023a), using their original source code. For Latent SDE, we formulate adjoint SDE proposed by Li et al. (2020) and train backpropagation incorporated with the original KL divergence.

To ensure a fair comparison, this study used the original architecture across all methods. However, because the optimal hyperparameters vary between the methods and data, this study used the Python library `ray`[20] (Moritz et al., 2018; Liaw et al., 2018) for hyperparameter tuning. With the use of this library, hyperparameters are optimally tuned to minimize validation loss automatically, contrasting with previous research that required manual tuning for each dataset and model. In our experiments, we identified the optimal hyperparameters for regular time series and applied them to irregular time series. For all methods, we employed the (explicit) Euler method as the ODE solver.

In each model and dataset, the following hyperparameters are optimized to minimize validation loss: hidden vector dimensions $n_h$, and the number of layers $n_l$. For RNN-based methods such as RNN, LSTM, and GRU, the fully-connected layer employs hyperparameters. For NDE-based methods, they are utilized to embed layer and vector fields. The hyperparameters are optimized as follows: the learning rate $lr$ from $10^{-4}$ to $10^{-1}$ using log-uniform search; $n_l$ from $\{1, 2, 3, 4\}$ using grid search; and $n_h$ from $\{16, 32, 64, 128\}$ using grid search. The batch size was selected from $\{16, 32, 64, 128\}$ with respect to total data size. All models were trained for 100 epochs, and the best model was

---

[12]https://github.com/patrick-kidger/torchcde
[13]https://github.com/google-research/torchsde
[14]https://github.com/patrick-kidger/NeuralCDE
[15]https://github.com/mlech26l/ode-lstms
[16]https://github.com/jambo6/neuralRDEs
[17]https://github.com/sheoyon-jhin/ANCDE
[18]https://github.com/sheoyon-jhin/EXIT
[19]https://github.com/alflsowl12/LEAP
[20]https://github.com/ray-project/ray

selected based on the lowest validation loss, ensuring that the chosen model generalizes well to unseen data. These design choices and training strategies aim to create a robust, well-performing model that effectively addresses the given classification tasks.

# F    DETAILED RESULTS OF 'ROBUSTNESS TO MISSING DATA' EXPERIMENTS

Our analysis involved a thorough evaluation of 30 datasets, considering various scenarios such as regular time series and time series with missing data rates of 30%, 50%, and 70%. Also, we considered different characteristics of time series, such as univariate or multivariate.

We tested Neural CDE using various controlled paths in Table 16: linear, rectilinear, natural cubic spline, and hermite cubic splines. Hermite cubic spline demonstrated superior performance compared to the Neural CDE with other control paths, leading us to select it for our proposed methods.

Table 16 presents an ablation study conducted to assess the performance of the proposed methods: Neural LSDE, Neural LNSDE, and Neural GSDE. It also compares the naïve Neural SDE with various components. As expected, the careful design of Neural SDEs can improve the classification performance. Also, we can observe that incorporating the control path into the model leads to a significant improvement in classification performance. Furthermore, employing a nonlinear neural network for the diffusion term further enhances the performance compared to using an affine transformation.

Table 16: Comparison of average accuracy and average cross-entropy loss from the ablation study. For Neural CDEs, different controlled paths were examined: (l)inear, (r)ectilinear, (n)atural cubic spline, and (h)ermite cubic splines. For Neural SDEs, $\zeta$ indicates whether the drift function incorporates the controlled path or not. 'N' or 'L' denote the architecture of diffusion function $\sigma$ with an affine function or a nonlinear neural network. **The best** and the second best are highlighted.)

| Methods | $\zeta$ | $\sigma$ | Univariate datasets (15) | | Multivariate datasets (15) | | All datasets (30) | |
|---|---|---|---|---|---|---|---|---|
| | | | Accuracy | Loss | Accuracy | Loss | Accuracy | Loss |
| Neural CDE (l) | | | 0.593 (0.094) | 0.809 (0.107) | 0.770 (0.037) | 0.595 (0.086) | 0.682 (0.065) | 0.702 (0.097) |
| Neural CDE (r) | – | | 0.550 (0.096) | 0.879 (0.090) | 0.716 (0.045) | 0.715 (0.105) | 0.633 (0.071) | 0.797 (0.098) |
| Neural CDE (n) | | | 0.597 (0.083) | 0.795 (0.084) | 0.773 (0.042) | 0.590 (0.095) | 0.685 (0.063) | 0.692 (0.089) |
| Neural CDE (h) | | | 0.611 (0.092) | 0.804 (0.119) | 0.777 (0.044) | **0.549 (0.101)** | 0.694 (0.068) | 0.676 (0.110) |
| **Neural SDE** | O | N | 0.615 (0.090) | 0.736 (0.089) | 0.760 (0.036) | 0.618 (0.092) | 0.688 (0.063) | 0.677 (0.091) |
| | O | L | 0.535 (0.095) | 0.838 (0.094) | 0.752 (0.040) | 0.633 (0.087) | 0.643 (0.068) | 0.736 (0.090) |
| | X | N | 0.516 (0.084) | 0.914 (0.069) | 0.515 (0.045) | 1.248 (0.085) | 0.516 (0.064) | 1.081 (0.077) |
| | X | L | 0.498 (0.087) | 0.929 (0.078) | 0.514 (0.050) | 1.255 (0.081) | 0.506 (0.068) | 1.092 (0.080) |
| **Neural LSDE** | O | N | 0.604 (0.081) | 0.752 (0.084) | **0.783 (0.031)** | 0.572 (0.080) | 0.694 (0.056) | 0.662 (0.082) |
| | O | L | 0.533 (0.089) | 0.877 (0.086) | 0.745 (0.038) | 0.668 (0.085) | 0.639 (0.064) | 0.772 (0.085) |
| | X | N | 0.530 (0.069) | 0.856 (0.060) | 0.527 (0.054) | 1.177 (0.082) | 0.528 (0.060) | 1.039 (0.074) |
| | X | L | 0.505 (0.064) | 0.912 (0.061) | 0.518 (0.057) | 1.210 (0.101) | 0.512 (0.059) | 1.083 (0.088) |
| **Neural LNSDE** | O | N | **0.654 (0.073)** | **0.701 (0.091)** | 0.780 (0.029) | 0.577 (0.070) | **0.717 (0.051)** | **0.639 (0.080)** |
| | O | L | 0.586 (0.087) | 0.765 (0.083) | 0.764 (0.044) | 0.617 (0.108) | 0.675 (0.066) | 0.691 (0.096) |
| | X | N | 0.532 (0.077) | 0.878 (0.060) | 0.502 (0.051) | 1.254 (0.082) | 0.517 (0.064) | 1.066 (0.071) |
| | X | L | 0.528 (0.085) | 0.890 (0.069) | 0.510 (0.047) | 1.257 (0.089) | 0.519 (0.066) | 1.074 (0.079) |
| **Neural GSDE** | O | N | 0.633 (0.091) | 0.742 (0.086) | 0.772 (0.036) | 0.598 (0.077) | 0.703 (0.063) | 0.670 (0.081) |
| | O | L | 0.572 (0.083) | 0.796 (0.084) | 0.748 (0.039) | 0.653 (0.094) | 0.660 (0.061) | 0.724 (0.089) |
| | X | N | 0.531 (0.074) | 0.868 (0.052) | 0.510 (0.045) | 1.261 (0.093) | 0.520 (0.060) | 1.064 (0.072) |
| | X | L | 0.525 (0.078) | 0.893 (0.072) | 0.509 (0.052) | 1.261 (0.093) | 0.517 (0.065) | 1.077 (0.083) |

Figures 3 and 4 display the classification outcomes for all datasets over the four missing rate scenarios. Each point in figure showcases the average classification score for every dataset and missing rate per method. The results highlight the variability in performance based on the dataset and its specific characteristics, emphasizing the importance of selecting an appropriate model that suits the specific properties of the time series data. These characteristics may include the degree of missing data, the presence of noise, the complexity of patterns, and the distribution of the data value.

In conclusion, it is vital to consider the inherent properties of the time series data when choosing the most appropriate model for a given dataset. By doing so, we can develop robust and accurate models that address the specific challenges of diverse time series datasets, resulting in better generalization and performance across various applications and domains.

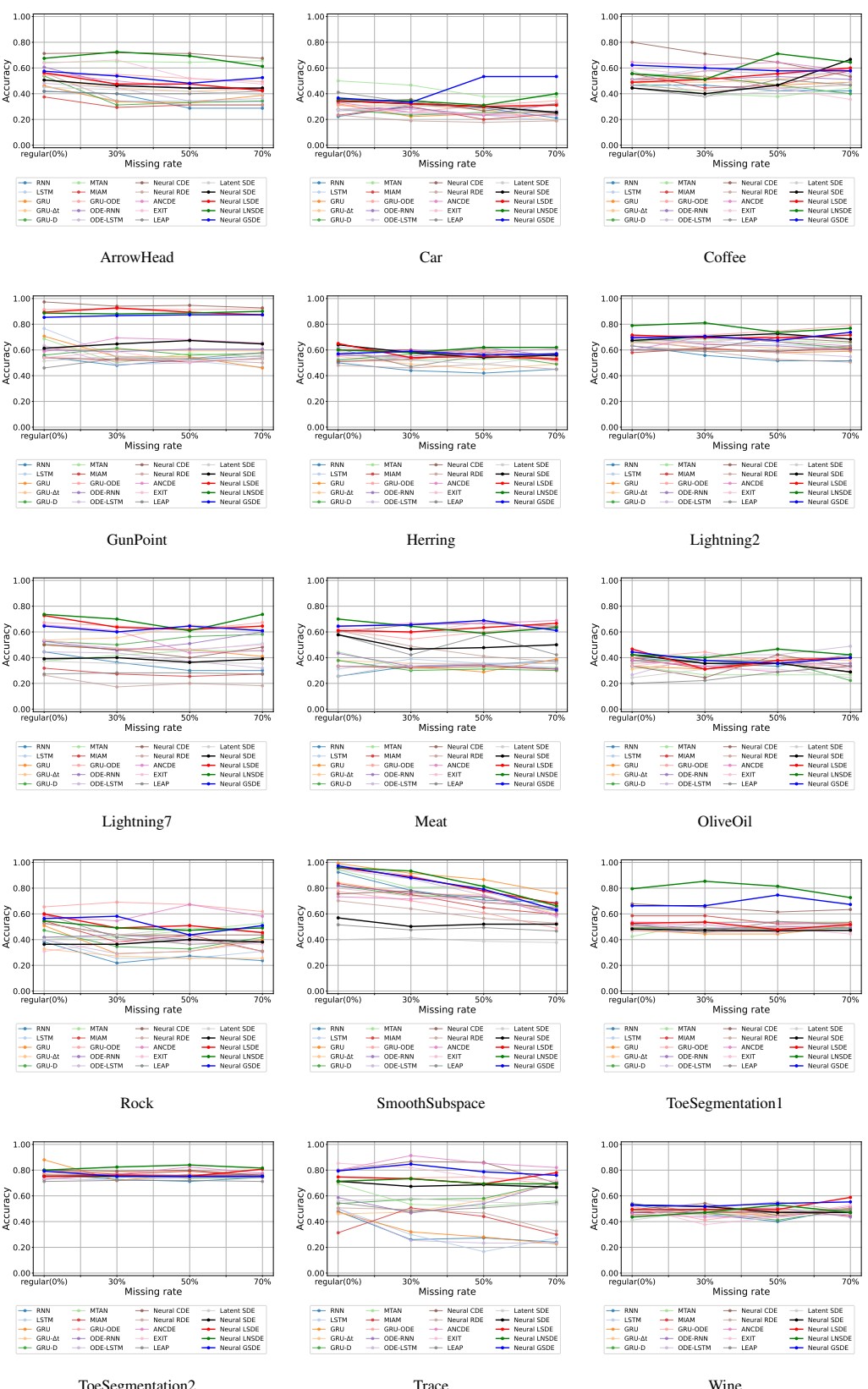

Figure 3: Classification result of the 15 univariate datasets with all four settings

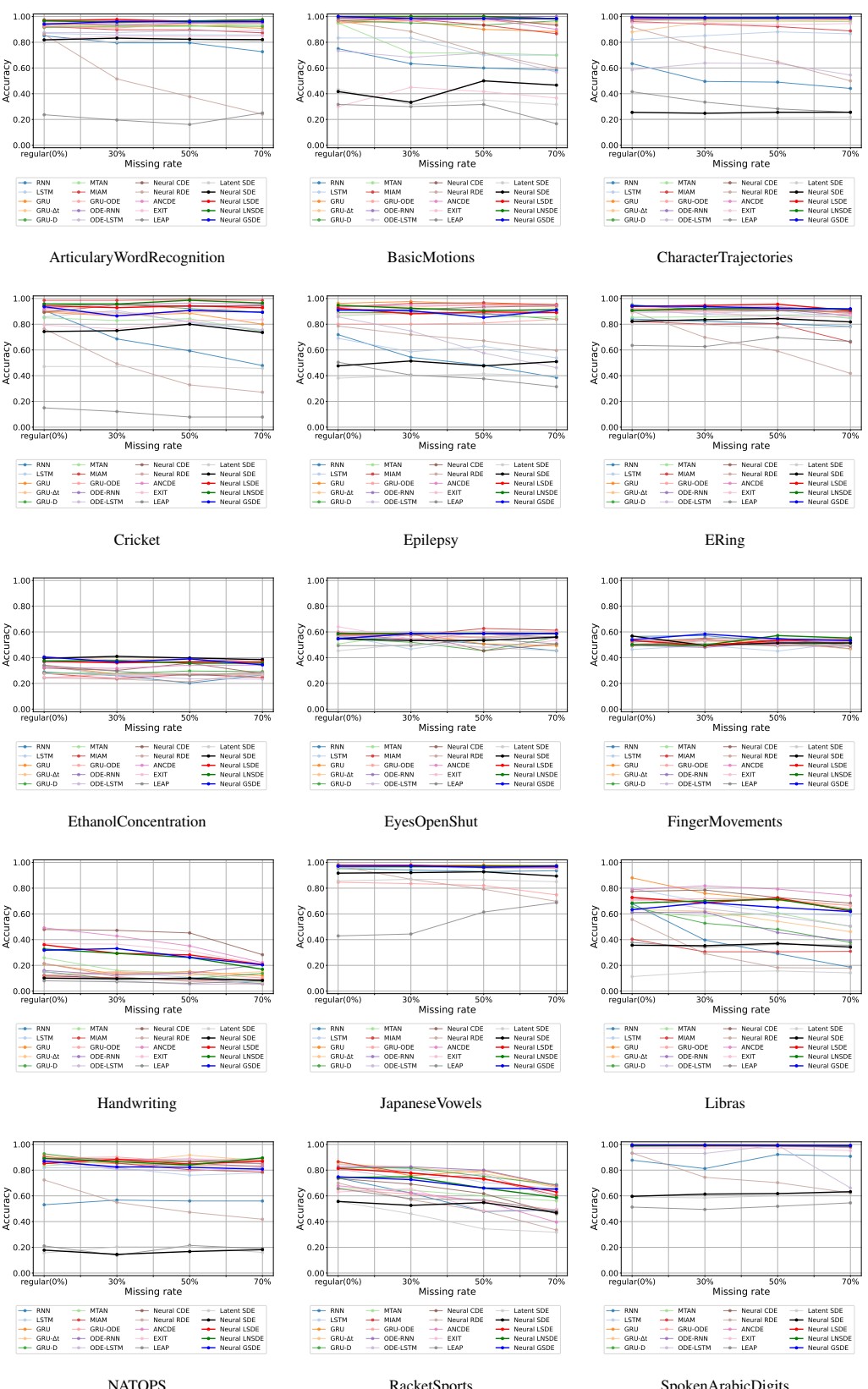

Figure 4: Classification result of the 15 multivariate datasets with all four settings

