# OpenReview forum: "Stable Neural Stochastic Differential Equations in Analyzing Irregular Time Series Data"
_ICLR.cc/2024/Conference — ICLR 2024 spotlight_

### Official Review · Reviewer_gG9F · 2023-10-31

**Soundness:** 3 good
**Presentation:** 3 good
**Contribution:** 3 good
**Rating:** 8
**Confidence:** 4

**Summary:**

The problem of learning neural stochastic differential equations (NSDE) to solve classification/interpolation tasks in the context of (irregularly sampled) time series data is considered. The authors focus on the analysis of theoretically well-defined SDE classes that exhibit desirable properties in terms of parameterization of drift and diffusion coefficients by neural networks, e.g., in so-called NSDEs. In contrast to naive NSDEs, which can (theoretically) learn almost arbitrary classes of functions, the authors restrict themselves to classes of SDEs for which (i) a (uniquely) strong solution exists, (ii) which can be approximated in a numerically stable manner, and (iii) which remain robust to input perturbations. In addition, they build on concepts from the field of controlled differential equations, which are known to improve model performance on irregularly sampled time series data.  Extensive experiments with established benchmark data sets are included. These provide empirical evidence for the proposed improvements.

**Strengths:**

I appreciate the idea of taking a step back from the state of unlimited expressiveness in NSDE and instead concentrating on sub-classes of SDEs that have favorable properties that combine well with the functional class properties of neural networks. The authors reveal shortcomings associated with the use of unrestricted parameterization of drift and diffusion coefficients by standard neural networks. In turn, an ablation study empirically supports the assumption that careful design of drift and diffusion coefficients is indeed reflected in improved model performance.
The content of the paper is well organized, original to the best of my knowledge and shows no obvious spelling or grammatical flaws.
Last but not least, I enjoyed the theoretically analysis of robustness under distribution shift.

**Weaknesses:**

1. I don't see the necessity to include the details on Neural ODE and CDE into the main manuscript. However, thats more ore less a
matter of taste.
2. (Section 4.2) Comparing such a rich variety of models is challenging. The main difficulty arises due to major differences in model structure. E.g., vanilla RNN based methods are by nature not capable to process irregularly sampled time series. As reported, data imputation strategies must be applied additionally. Another challenge arises from comparing methods that include a control mechanism (e.g., Neural CDE) with methods that do not (e.g., NSDE). The former are able to continuously correct the sampled trajectories over time during learning, while the latter can largely only adjust the initial state. However, the authors elaborate on the latter problem in Table 11, where the proposed methods nevertheless showed their advantage. However, I can't escape the impression that the built-in control mechanism is a big part of the success; because Neural CDE often takes the closely followed second place.

Nevertheless, I am on the positive side of this work.

**Questions:**

1. For example, to evaluate robustness empirically, *explicit Euler* is used for all experiments (see page 25). What are the reasons for this choice? I am very curious about the impact of different numerical solution methods on these and other results. Can you explain this in more detail?
2. Are you planning to release your Code which would unlock reproducibility of the results?

Minor:

3. Aren't the initial state in the Eq. (2) and (3) supposed to be vectors and therefore should be bold?

**Details Of Ethics Concerns:**

--

---

> ### Author Response · Authors · 2023-11-20
> **Response to Weaknesses and Questions**
>
> >**Response to weakness 1**
>
> Thank you for your valuable suggestion. After careful consideration, we have decided to keep them as they are.
>
> >**Response to weakness 2**
>
> Thank you for the insightful comment. Vanilla RNN-based methods face challenges in handling irregularly sampled time series, necessitating additional efforts like data imputation. Other traditional methods also struggle with irregularly sampled time series, requiring extra inputs such as observation intensity, masks for missing data, or specialized architectural features. NDEs without controlled paths heavily rely on initial states, which limits their applicability and performance.
>
> These limitations have motivated us to incorporate the controlled path, which significantly enhances the performance of models in handling irregularly sampled time series. However, as shown in **Table 11** of the *original manuscript* (**Table 16** of the *revised manuscript*), the naive Neural SDE combined with the controlled path does not outperform the Neural CDE. Therefore, we would like to highlight the importance of well-designed drift and diffusion functions, along with the controlled path, in improving model performance.
>
> >**Response to question 1**
>
> Thank you for your insightful comments. In our numerical experiments, we utilized three numerical solvers: the Euler-Maruyama, Milstein, and Stochastic Runge-Kutta (SRK) methods, provided by the $\texttt{torchsde}$ Python library. The Euler-Maruyama method, an explicit forward method, is highly efficient for solving high-dimensional SDEs, making it a preferable choice over other implicit and high-order solvers like the Milstein and SRK methods. The **Table 8** in the *revised manuscript* presents a comparison of training accuracy and runtime across these three solvers on the `BasicMotion' dataset with a 50% missing rate. This comparison highlights the exceptional efficiency of the Euler-Maruyama method. Consequently, we chose the Euler-Maruyama method for computing the numerical solutions of the Neural SDEs in all our experiments.
>
> >**Response to question 2**
>
> Here is the link for the code: https://bit.ly/3XCKiN5. The link for the implementation code can be found in Section 4.1 - Experimental Protocols of the original manuscript. This section provides detailed information about the experimental setup and the corresponding implementation for each experiment. Also, full details are explained in Appendix D.
>
> >**Response to question 3**
>
> Yes, they should be written in bold. We have now updated the manuscript accordingly.

---

> > ### Comment · Reviewer_gG9F · 2023-11-21
> >
> > I thank the authors very much for their response and for the discussion
> > of my questions and concerns.
> >
> > You have clarified things, and I will increase my score correspondingly, under the assumption that these clarifications will make it into the updated version of the paper.
> >
> > Thank you again.

---

### Official Review · Reviewer_Q7tH · 2023-10-31

**Soundness:** 3 good
**Presentation:** 3 good
**Contribution:** 3 good
**Rating:** 6
**Confidence:** 3

**Summary:**

This paper introduces three stable classes of Neural SDEs (Langevin-type SDE, Linear Noise SDE, and Geometric SDE) to capture complex dynamics and improve robustness under distribution shifts in time series data. Theoretically, this paper shows the existence and uniqueness of the solutions of these SDEs, and presents their performance guarantee under distribution shifts. Extensive experiments are conducted to validate the good performance.

**Strengths:**

Topic-wise, modeling time series data with irregular sampling intervals and missing values is an essential research topic and of great importance in practice.

Theory-wise, this paper proves the existence and uniqueness of the solutions of the proposed three SDEs, and shows their robustness to input data.

Additionally, extensive numerical results are presented to compare the proposed method with existing algorithms for time series modeling.

**Weaknesses:**

The computational complexity of the proposed method is certainly high.

This paper lacks sufficient details on the implementation of the method, especially when there are irregular time steps and missing data in the series.

For time series data, in addition to interpolation and classification tasks, it would also be meaningful to consider forecasting tasks as well, which seems to be absent in this paper.

**Questions:**

Detailed discussions on the training procedure are needed, especially for dealing with irregular time steps and missing data. For instance, with different irregular time steps across different sample time series, is it still possible to train the algorithm using mini-batch optimization; and is there a way to improve the computational efficiency in practice? When there is missing data in the sequence, how do we deal with missing values in the training phase -- are these missing values being imputed or ignored during the pre-processing step?

More explanations are needed for Figure 1 — line (i) exhibits a constant loss, and in fact, most of the loss trajectories are not satisfactory, with unstable trends and not decaying with the increase of epochs.

From Table 5, it is a bit confusing why there is no result for LSDE, LNSDE, and GSDE-‘+Z’.

The downstream tasks considered in this paper are interpolation and classification; it could also be meaningful and worthwhile to consider the prediction task for time series data as well.

---

> ### Author Response · Authors · 2023-11-20
> **Response to Weaknesses and Questions**
>
> > **W1: The computational complexity of the proposed method is certainly high.**
>
> We have conducted additional experiments on the `BasicMotion' dataset with 50\% dataset to compare the training time of the proposed Neural SDEs with other methods. The results, as shown in **Table 9** of the *revised manuscript*, indicate that it takes slightly longer for the proposed Neural SDEs to be trained for 100 epochs than the naive Neural SDEs. However, it is crucial to observe that they train faster and achieve the best accuracy compared to other neural differential equations such as Latent SDE, LEAP, EXIT, ANCDE, Neural CDE, ODE, RNN, GRU ODE, and Neural RDE. Therefore, we emphasize the benefits of the proposed Neural SDEs, highlighting their powerful empirical performance along with theoretical advantages like robustness and well-posedness.
>
> > **W2: This paper lacks sufficient details on the implementation of the method~**
>
> This part will be addressed in the response to **Q1**.
>
> > **W3: For time series data, in addition to interpolation and classification tasks~**
>
> This part will be addressed in the response to **Q4**.
>
> > **Q1: Detailed discussions on the training procedure**
>
> We thank the referee for the comments.
> - **Is it still possible to train the algorithm using mini-batch optimization?**
> YES, we employ the `torchsde` library, which leverages the adjoint-sensitivity algorithm [1] to train Neural SDEs. The adjoint sensitivity algorithm is compatible with the mini-batch scheme, allowing us to efficiently compute gradients of the Neural SDEs even in the presence of irregular time steps.
>
> - **Is there a way to improve the computational efficiency in practice?**
> To improve the computational efficiency of Neural SDEs in practice, we recommend some simple and effective methods: 1) use the Euler-Maruyama solver for solving SDEs, see **Table 8** of the revised manuscript for the computational efficiency of the Euler-Maruyama solver, 2) employ the adjoint sensitivity algorithm for computing gradients, and 3) use tanh activation functions at the last layer.
>
> - **how do we deal with missing values in the training phase?**
> Interpolation is applied for the given data with irregularly-sampled or missingness. Then, the interpolated value is used to map into the initial value of the Neural SDEs. This approach aligns with the methods used by Kidger et al. [2] and subsequent works. Specifically, we used hermite cubic splines with backward differences [3].
>
> > **Q2: More explanations are needed for Figure 1**
>
> Thank you very much for your valuable comment. The purpose of this experiment was to investigate the impact of the diffusion term on the performance of Neural SDEs. Therefore, we only implemented naive SDEs with various diffusion functions, but without a controlled path and a properly defined drift function. Consequently, the overall loss may not be as optimal as in our final model. This highlights the importance of well-designed drift and diffusion functions, along with the incorporation of a controlled path, which are key contributions of our work.
>
> > **Q3: From Table 5, it is a bit confusing why there is no result for LSDE, LNSDE, and GSDE-‘+Z’.**
>
> Thank you for your comment. In **Table 5** of the *revised manuscript*, we have reorganized the table based on whether to incorporate the control path and whether to use the diffusion function as an affine function or a nonlinear neural network, to reduce confusion. The results clearly show that, compared to naive Neural SDEs, our proposed neural SDEs with well-designed diffusion and drift functions, along with the controlled path, significantly enhance performance.
>
> > **Q4: The downstream tasks considered in this paper~**
>
> We conducted additional experiments to evaluate the performance of the proposed methods in forecasting tasks. For these experiments, we utilized the MuJoCo dataset [4], and a detailed description of the data is provided in Appendix C of the revised manuscript. **Table 12** in Appendix E of the *revised manuscript* presents the forecasting performance across varying data drop ratios. We confirmed that our methods consistently demonstrate lower MSE scores, indicating their superior forecasting capabilities.
>
>
> **References**
> - [1] Li, X., Wong, T. K. L., Chen, R. T., & Duvenaud, D. (2020, June). Scalable gradients for stochastic differential equations. In International Conference on Artificial Intelligence and Statistics (pp. 3870-3882). PMLR.
> - [2] Kidger, P., Morrill, J., Foster, J., & Lyons, T. (2020). Neural controlled differential equations for irregular time series. Advances in Neural Information Processing Systems, 33, 6696-6707.
> - [3] Morrill, J., Kidger, P., Yang, L., & Lyons, T. (2021). Neural controlled differential equations for online prediction tasks. *arXiv preprint arXiv:2106.11028*.
> - [4] Tassa, Y., Doron, Y., Muldal, A., Erez, T., Li, Y., Casas, D. D. L., ... & Riedmiller, M. (2018). Deepmind control suite. arXiv preprint arXiv:1801.00690.

---

> > ### Comment · Reviewer_Q7tH · 2023-11-23
> >
> > I would like to thank the authors for their kind response. My overall rating remains.

---

### Official Review · Reviewer_8YRs · 2023-11-01

**Soundness:** 3 good
**Presentation:** 3 good
**Contribution:** 3 good
**Rating:** 6
**Confidence:** 3

**Summary:**

This paper addresses the challenges posed by irregular sampling intervals and missing values in real-world time series data. The authors propose three classes of Neural Stochastic Differential Equations (Neural SDEs) to improve robustness under distribution shifts in time series data. The proposed Neural SDEs include Langevin-type SDE, Linear Noise SDE, and Geometric SDE. The study demonstrates the robustness of these Neural SDEs theoretically and through extensive experiments, showing their effectiveness in handling real-world irregular time series data and maintaining excellent performance under distribution shifts due to missing data.

**Strengths:**

The authors provide a solid theoretical foundation for the proposed Neural SDEs, including the existence and uniqueness of solutions.

The robustness section in the paper provides valuable insights into the proposed Neural SDEs' resilience against distribution shifts and input perturbations.

 The paper conducts extensive experiments to validate the effectiveness of the proposed Neural SDEs. The models are tested on various datasets, and their robustness is analyzed under different missing rates, providing a comprehensive evaluation.

**Weaknesses:**

I didn’t identify major weakness in this paper; however, there are a few minor concerns that I would like to address:

1. Section 3.4 could be more explicit in detailing how the controlled path is incorporated into the Neural SDEs. More comprehensive explanations or illustrations could help in understanding the model’s architecture and functionality better.

2.  While section 3.3 discusses the robustness of the proposed Neural SDEs under distribution shifts, it might benefit from a more thorough exploration or comparison with other neural SDEs' robustness aspects in related works.

3. What is the $\| \sigma_\theta  \|$ in Theorem 3.6 ?

4. In the experiments of missing data (Table 4), the proposed Neural SDEs show only marginal improvements when compared to the Neural CDE model. How do the theoretical bounds in Theorems 3.5 and 3.6 relate to the robustness of the Neural SDEs in practical implementations? Are these bounds tight or rather loose in actual application scenarios?

5. The implementation code is not provided.

**Questions:**

The details of the proposed models seem unclear.  Can you clarify which neural networks are used in the diffusion and drift functions of each Neural SDE?

Additionally,  how are the proposed neural SDEs solved, especially the neural GSDE?   Do you employ numerical solvers for the Neural SDEs? If so, which specific solver was utilized, and was there any ablation study conducted to evaluate its effectiveness?

How much computational time is required for training the proposed Neural SDEs, and what is the complexity of these models?

---

> ### Author Response · Authors · 2023-11-20
> **Response to Weaknesses**
>
> > **Response to weakness 1**
>
> Thank you very much for your comment. Due to the page limit, we deferred the detailed explanation of how the controlled path is integrated into our Neural SDEs to Appendix A.1 of the original manuscript. It contains the associated SDEs combined with the controlled path, e.g., see Eqs (12), (13), (14), and discusses that the drift function with the controlled path also satisfies the Lipschitz continuity condition.
>
> At the implementation stage, we first consider a neural network $\zeta: \mathbb R_+ \times \mathbb{R}^{d_z} \times \mathbb{R}^{d_x} \rightarrow \mathbb{R}^{d_z}$ by concatenating the latent variable $\mathbf{z}(t)$ and the controlled path $X(t)$ to produce $\overline{\mathbf{z}}(t)$. Then, the drift functions of the proposed Neural SDEs take $\overline{\mathbf{z}}(t)$ instead of $\mathbf{z}(t)$, effectively incorporating the controlled path.
>
> > **Response to weakness 2**
>
> Thank you for your valuable suggestion. As injected noise in SDEs can destabilize a stable system, the performance of naïve neural SDEs can be extremely vulnerable. For example, we found that the naïve Neural SDE was not properly trained and performed poorly in the interpolation and classification experiments of Section 4.1. Furthermore, in GSDE, it is crucial to satisfy the uniformly bounded condition as stated in Eq (7). This is achieved by assuming that neural networks for the drift and diffusion functions use sigmoid or tanh activation functions at their last layers. Without such careful consideration, GSDE also fails to achieve the robustness under distribution shifts described in Theorem 3.6.
>
> In addition, we would like to highlight that, to the best of our knowledge, our work is the first attempt to rigorously study the robustness of Neural SDEs under distribution shifts. The most closely related work is [1], which reiterates the almost sure exponential stability result found in the textbook [2]. However, [1] focuses only on the LNSDE type and does not clearly discuss the relationship between the stability of SDEs and the robustness of trained Neural SDEs.
>
>
> > **Response to weakness 3**
>
> We have now provided a clear definition of $\sigma_\theta$ on page 5 in the revised manuscript as follows:
>
> ''For our stability analysis, we assume $\sigma(t;\theta_\sigma)$ to be either a constant $\sigma_{\theta}$ or to have a limit such that
> $\lim_{t\rightarrow \infty}\sigma(t;\theta_\sigma) =: \sigma_{\theta}.$"
>
> > **Response to weakness 4**
>
> We thank the referee for the comments. Although our numerical results in Table 4 indicate that the Neural CDE model seems quite robust with respect to missing data, potentially due to the good properties of controlled paths such as smoothness and boundedness discussed in [3], a rigorous analysis of the robustness of the Neural CDE under distribution shifts remains unaddressed in the literature.
>
> Theorems 3.5 and 3.6 provide non-asymptotic upper bounds for the differences between the output distributions of the original input data and its perturbed version in terms of the degree of distribution shift $\rho$ and the depth of the Neural SDEs $T$. In practical implementation, the key implication of Theorems 3.5 and 3.6 is that smaller perturbations and larger depths yield smaller differences. Therefore, it is expected that a larger $T$ will yield a model more robust to missingness and irregularity in data. To confirm this, we have conducted an additional experiment to investigate the performance changes in Neural SDEs with respect to $T$. Please also refer to **Table 7** in the *revised manuscript* in detail.
>
> Regarding the tightness of the bounds, they also depend on the Lipscthiz constant $L_F$ for the classifier $F$, the initial condition $L_h$, and the contraction parameters $c_1$ and $c_2$. Notably, $c_1$ and $c_2$ are closely linked to the dimension of the given SDE. Thus, the tightness of the bounds can vary significantly depending on the problems. In particular, our assumptions do not impose convexity on the drift and diffusion functions, meaning our theoretical results encompass pathological cases. Therefore, the upper bounds may be loose in some cases.
>
> > **Response to weakness 5**
>
> Here is the link to the code: https://bit.ly/3XCKiN5. The implementation code can be found in Section 4.1 of the original manuscript. For comprehensive details of the experimental setup, please refer to Appendix D.
>
> **References**
> - [1] X. Liu et al. Neural SDE: Stabilizing Neural ODE networks with stochastic noise. arXiv, 2019.
> - [2] X. Mao. Stochastic differential equations and applications. Elsevier, 2007.
> - [3] J. Morrill et al. Neural Controlled Differential Equations for Online Prediction Tasks, 2021, arxiv.

---

> ### Author Response · Authors · 2023-11-20
> **Response to Questions**
>
> > **Response to question about the clarification of neural networks**
>
> We employ standard feedforward neural networks, commonly known as multilayer perceptron models, for both the drift and diffusion functions in our Neural SDEs. These networks vary in the number of layers $n_l$ and the number of neurons per layer $n_h$. To determine the optimal hyperparameters, we tested combinations of $n_l=[1,2,3,4]$ and $ n_h=[16,32,64,128]$ across all datasets. For the diffusion term, we also evaluated the effectiveness of linear affine functions. Detailed results can be found in the updated **Table 5**. Additionally, we apply tanh functions at the final layer as recommended by [4]. For a comprehensive overview of our training strategy and model architectures, please refer to Appendix D.1.
>
> > **Response to question about the numerical solvers for the Neural SDEs**
>
> Thank you for your insightful comments. In our numerical experiments, we utilized three numerical solvers: the Euler-Maruyama, Milstein, and Stochastic Runge-Kutta (SRK) methods, provided by the torchsde Python library. Each solver has its own strengths and weaknesses in terms of computation speed and convergence rate. The Euler-Maruyama method, an explicit forward method, is notably efficient for solving high-dimensional SDEs. While other implicit and high-order solvers like the Milstein and SRK methods offer faster convergence, they demand more computational time.
>
> In particular, as our proposed Neural SDEs are well-posed and achieve the best convergence rate (0.5), the Euler-Maruyama method offers significantly faster training times compared to other methods without losing noticeable accuracy. The **Table 8** in the *revised manuscript* shows a comparison of training accuracy and runtime across these three solvers on the `BassicMotion' dataset with 50\% missing rate, highlighting the exceptional efficiency of the Euler-Maruyama method. Therefore, we chose the Euler-Maruyama method for computing the numerical solutions of the Neural SDEs in all our experiments.
>
> > **Response to question about the computational time**
>
> We thank the referee for the comment. We have conducted additional experiments on the `BasicMotion' dataset with 50\% dataset to compare the training time of the proposed Neural SDEs with other methods. The results, as shown in the **Table 9** of the *revised manuscript*, indicate that it takes slightly longer for the proposed Neural SDEs to be trained for 100 epochs than the naive Neural SDEs. However, it is crucial to observe that they train faster and achieve the best accuracy compared to other neural differential equations such as Latent SDE, LEAP, EXIT, ANCDE, Neural CDE, ODE, RNN, GRU ODE, and Neural RDE. Therefore, we emphasize the benefits of the proposed Neural SDEs, highlighting their powerful empirical performance along with theoretical advantages like robustness and well-posedness.
>
> **References**
> - [4] Kidger, P., Morrill, J., Foster, J., & Lyons, T. (2020). Neural controlled differential equations for irregular time series. Advances in Neural Information Processing Systems, 33, 6696-6707.

---

> > ### Comment · Reviewer_8YRs · 2023-11-23
> >
> > Appreciating the authors' response, I have decided to maintain my current score.

---

### Author Response · Authors · 2023-11-20
**Global Response to All Reviewers**

The authors appreciate the recognition of the value and significance of our paper and express gratitude to the valuable comments provided by the reviewers. We have addressed each weakness and question raised by the reviewers, and the revised manuscript, with modifications highlighted in blue, has been uploaded in response.

In the rebuttal process, we conducted additional experiments as outlined below to enhance the completeness of the paper.

1. Ablation study for drift function and diffusion function (**Tables 5, 16**) (_in response to Reviewer Q7tH_)
2. Forecasting task (**Tables 11, 12**)  (_in response to Reviewer Q7tH_)
3. Sensitivity analysis for  Neural SDEs' depth $T$ (**Table 7**) (_in response to Reviewer 8YRs_)
4. Computation time comparison study (**Table 9**) (_in response to Reviewer 8YRs, Q7tH_)
5. Numerical solver comparison study (**Table 8**) (_in response to Reviewer 8YRs, Q7tH, gG9F_)

$*$ The table numbers mentioned above are based on the revised manuscript.

---

### Author Response · Authors · 2023-11-22
**A Gentle Reminder**

We appreciate your acknowledgment of the value and significance of our paper, and we are grateful for the valuable comments provided by the reviewers. We have responded to each weakness and question raised by the reviewers, and the revised manuscript has been uploaded with modifications highlighted in blue.

We would like to express our sincere gratitude to _Reviewer gG9F_ for reviewing our responses and increasing the score.

To _Reviewer 8YRs and Q7tH_, we kindly remind you that **the discussion period will end in less than a day**. We just wonder whether there is any further concern and hope to have a chance to respond before the discussion period ends.

Sincerely,

Authors

---

### Public Comment · ~YongKyung_Oh1 · 2024-05-20

Please refer the paper and code in the following links. Thank you for your interests.
- paper: https://arxiv.org/abs/2402.14989
- code: https://github.com/yongkyung-oh/Stable-Neural-SDEs

---

### Meta-Review · Area_Chair_iBmF · 2023-12-08

**Metareview:**

This paper proposes three parameterizations of Neural SDEs—Langevin-type SDE, Linear Noise SDE, and Geometric SDE with a view to improve the robustness of Neural SDEs to distribution shift. The paper shows the existence and uniqueness of the solutions of these SDEs. All reviewers found the paper well written, well motivated and overall as presenting a practical solution to a real problem in the context of Neural SDEs. I think this will make a good addition to the proceedings. The theory (bounding the difference between the output of the model under distribution shift and the depth of the SDE) is nice but the experimentation does a good job at studying these questions on a variety of different datasets.

Please do make the changes as per the rebuttal to the updated manuscript and also release a publicly repository for code to ensure reproducibility of your work.

**Justification For Why Not Higher Score:**

This does not introduce a new class of models but instead proposes clever, practical variations on existing models and studies their utility.

**Justification For Why Not Lower Score:**

It has a nice mix of theory and experimentation that reviewers appreciated that goes beyond what might be needed for a poster presentation.

---

### Decision · Program_Chairs · 2024-01-16

Accept (spotlight)